# Japanese encephalitis virus neuropenetrance is driven by mast cell chymase

Justin T. Hsieh [1], Abhay P.S. Rathore [1,4], Gayathri Soundarajan[1] & Ashley L. St. John [1,2,3]

Japanese encephalitis virus (JEV) is a leading cause of viral encephalitis. However, the mechanisms of JEV penetration of the blood-brain-barrier (BBB) remain poorly understood. Mast cells (MCs) are granulated innate immune sentinels located perivascularly, including at the BBB. Here we show that JEV activates MCs, leading to the release of granule-associated proteases in vivo. MC-deficient mice display reduced BBB permeability during JEV infection compared to congenic wild-type (WT) mice, indicating that enhanced vascular leakage in the brain during JEV infection is MC-dependent. Moreover, MCs promoted increased JEV infection in the central nervous system (CNS), enhanced neurological deficits, and reduced survival in vivo. Mechanistically, chymase, a MC-specific protease, enhances JEV-induced breakdown of the BBB and cleavage of tight-junction proteins. Chymase inhibition reversed BBB leakage, reduced brain infection and neurological deficits during JEV infection, and prolonged survival, suggesting chymase is a novel therapeutic target to prevent JEV encephalitis.

[1] Program in Emerging Infectious Diseases, Duke-National University of Singapore Medical School, Singapore 169857, Singapore. [2] Department of Microbiology and Immunology, Yong Loo Lin School of Medicine, National University of Singapore, Singapore 119228, Singapore. [3] Department of Pathology, Duke University Medical Center, Durham 27710 NC, USA. [4]Present address: Department of Pathology, Duke University Medical Center, Durham 27710 NC, USA. Correspondence and requests for materials should be addressed to A. J. (email: ashley.st.john@duke-nus.edu.sg)

Japanese encephalitis virus (JEV) causes ~68,000 cases of viral encephalitis in Asia each year[1]. It is a positive-sense, single-stranded RNA virus belonging to the family *Flaviviridae* that is spread by mosquitos[2]. As a neurotropic virus, it can effectively cross the blood–brain barrier (BBB) to cause acute encephalitis in humans. Of the patients afflicted with JEV encephalitis, 25–30% of cases are fatal and 50% result in permanent neuropsychiatric sequelae such as recurrent seizure, paralysis, and intellectual disability[3]. In the central nervous system (CNS), JEV primarily infects neurons[4], yet it also activates supporting glial cells such as microglia and exacerbates neuronal death[5–7]. While JEV causes high morbidity and mortality, there are currently no approved therapies for preventing the development of neurological symptoms[8,9]. Advances in JEV treatment or prevention are impeded by the fact that the mechanisms for neurotropism, BBB penetration, and neuroinflammation in JEV infection are poorly understood.

The mechanism through which JEV gains entry into the CNS has not been clearly defined. The CNS is considered immune privileged and is segregated from peripheral tissues by the tight barrier of the BBB, in part, to avoid infection[10]. A hematological route for JEV entry into the brain has been suggested based on the diffuse infection in both human and mouse postmortem brain samples[4,11]. Breakdown of tight junctions (TJs) between brain endothelial cells has been shown to occur and can potentially facilitate JEV entry into the brain, as supported by JEV-induced breakdown of TJ proteins in vivo in mice[12]. Furthermore, the breakdown was driven by inflammatory cytokines and chemokines; however, the source of these factors remains to be identified. Although suspected to have an immune component, the factors initiating BBB breakdown during JEV infection remain elusive.

Mast cells (MCs) are one of the two types of resident immune cells in the CNS (the other type being microglia) and are strategically located near the BBB and the neurovascular unit, which includes brain endothelial cells, pericytes, astrocytes, microglia, and neurons[13,14]. MCs are of the hematopoietic myeloid lineage and act as innate immune sentinels for pathogens in peripheral connective and mucosal tissues[15,16]. Once activated by certain pathogens, MCs release pre-formed granules containing inflammatory mediators, vasoactive molecules, and proteases (including MC-specific proteases chymase and tryptase)[17]. Additionally, pathogen recognition by MCs leads to production of cytokines, chemokines, and eicosanoids de novo[15,18]. Together, these MC products are pro-inflammatory, vasoactive, and can mobilize other innate and adaptive immune cell types for optimal clearance of the pathogen.

MC responses to infection are often protective; however, recent evidence suggests that they could be detrimental in some circumstances. In the context of another Flaviviral pathogen, dengue virus (DENV), MCs induce significant vascular leakage and edema in peripheral tissues in response to the infection[19]. This was shown in mice to be the consequence of MC products acting on the endothelial cells of nearby blood vessels, leading to increased vascular permeability, and data in humans support that there is a correlation between MC activation and severe disease[20,21]. MCs may also increase disease severity in several sterile neuroinflammatory diseases, including multiple sclerosis, neuropathic pain, ischemic/hemorrhagic stroke, and traumatic brain injury[22–31]. In conditions such as ischemic-reperfusion injuries, traumatic brain injury, and stroke, the MC-specific protease chymase has been implicated in promoting BBB compromise, both directly and through regulating matrix metalloproteinase (MMP)-9 and -2. MMPs can also break down important TJ proteins such as zonula occludens (ZO)-1, ZO-2, claudin-5, and occludin[25,32–38].

While many studies of JEV have used attenuated strains for the purpose of vaccine development, much less is known regarding the mechanisms that regulate virulent JEV entry into the brain. Furthermore, the role of MCs as brain-resident immune cells during viral encephalitis is unknown. Using a genetic model of MC deficiency and a virulent clinical strain of JEV, we show that MCs trigger JEV-induced breakdown of the BBB, leading to worsened infection, morbidity, and mortality. Furthermore, we identified the MC-specific protease, chymase, as a key mediator of BBB compromise and JEV neuroinvasion that can be therapeutically targeted to reverse vascular leakage, limit infection and signs of disease, and prolong survival.

## Results

**JEV infection induces MC degranulation.** It is currently unknown if MCs in the brain are activated during viral encephalitis. To address this question, we examined the brains of JEV-infected animals for evidence of MC degranulation. Mice were infected with $2 \times 10^7$ plaque forming units (PFU) of JEV, strain Nakayama, which is highly virulent in mice and causes encephalitis[39–41]. In contrast to control tissues, where quiescent MCs could be observed within the brain, proximal to blood vessels, we observed degranulated MCs near brain endothelial cells of JEV-infected animals, 5 days post-infection, by toluidine blue staining (Fig. 1a, Supplementary Figure 1a and 1b). By a second method of immunofluorescence staining, we visualized activated MCs near the blood vessels in the brain parenchyma in areas where neighboring cells stained positive for JEV replication protein, nonstructural protein 3 (NS3) (Fig. 1b, Supplementary Figure 1c). Staining using a probe for the MC-granule-specific substance heparin showed punctate staining for released granules throughout the brain tissue (Fig. 1b). Inflammation in the brain, including infiltration of cells into the brain tissue surrounding blood vessels (Supplementary Figure 1d), was observed in histologic sections and a 90% mortality rate was observed when mice were infected peripherally with $2 \times 10^7$ PFU of Nakayama (Fig. 1c). These data show that MCs are activated and degranulate in the brain during lethal JEV encephalitis.

To determine whether JEV can induce degranulation of MCs directly without the influence of bystander brain cells, we exposed mouse bone marrow-derived MCs (BMMCs) to a live-attenuated strain of JEV (SA 14-14-2). In contrast to Nakayama, the vaccine strain SA 14-14-2 is not lethal in mice by peripheral injection, even at double the inoculating dose that was lethal for the Nakayama strain (Fig. 1c). The strain SA 14-14-2 was chosen for initial in vitro studies since it can be handled in a BSL2 setting and to assess whether JEV-induced degranulation occurs with multiple strains. At a multiplicity of infection (MOI) of 1, degranulation of BMMCs was observed by immunofluorescence staining (Fig. 1d). In untreated MCs, granules appeared tightly packed within the cell, but JEV treatment induced the release of granules that could be viewed on the slide outside of the tubulin-stained border of the cell (Fig. 1d). Quantification of the MC granule product β-hexosaminidase after exposure of MCs to both live (Fig. 1e) and UV-inactivated (Supplementary Figure 1e) JEV showed that JEV induced significant degranulation of BMMCs, which increased as the MOI increased. To corroborate that live-attenuated strain SA 14-14-2 also induced MC activation in vivo, serum concentrations of the MC-specific protease, chymase (also known as MCPT1), were measured in uninfected control (time = 0) and JEV-infected wild-type (WT) mice. Serum MCPT1 levels were elevated, particularly at earlier time points within 3 days of infection (Fig. 1f). Furthermore, similar to the more virulent strain (Fig. 1a), degranulating MCs could be identified in the brains of mice infected with SA 14-14-2 and

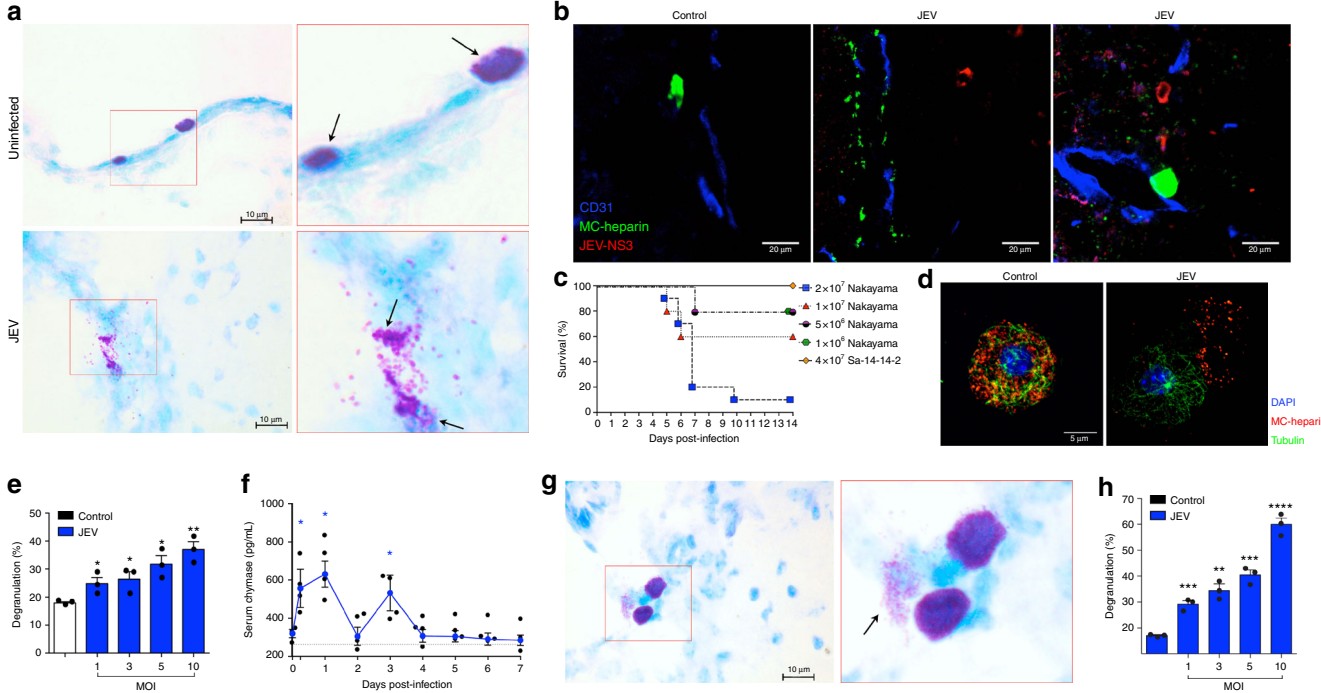

**Fig. 1** Mast cells degranulate in response to JEV. **a** WT mice infected with JEV displayed increased MC degranulation in the brain 5 days post-intra peritoneal (i.p.) infection with $2 \times 10^7$ plaque forming units (PFUs) of Nakayama JEV. Activated MCs were noted near brain blood vessels, as assessed by toluidine blue staining. Right images reveal details of boxed areas of the left images. Vehicle-treated brain sections contain well-granulated MCs while JEV-infected brain sections contain degranulated MCs (arrows). **b** MC degranulation was visualized near endothelial cells (blue; CD31) of the brain, with JEV infection (red; JEV NS3 protein) nearby, 5 days post-infection. MC granules are stained using the probe for heparin avidin-FITC (green). Vehicle-treated control brains showed well-granulated MCs. **c** i.p. Nakayama JEV infection caused inoculating titer-dependent mortality in WT mice ($n = 10$ for $2 \times 10^7$ group, $n = 5$ for all other groups). By 7 days, 90% mortality was induced by $2 \times 10^7$ PFU, therefore this titer was used for subsequent experiments. **d** JEV stimulation led to degranulation of mouse BMMCs, shown by immunofluorescence staining for MC granules (avidin-TRITC, red), tubulin (green), and DNA (blue). **e** Mouse BMMCs were activated in an MOI-dependent manner by JEV, as assessed by β-hexosaminidase assay. * denotes $P < 0.05$, ** denotes $P < 0.01$, determined by one-way ANOVA, compared to uninfected controls. **f** JEV infection increased serum chymase (MCPT1) concentrations in WT mice at 6 h, 1 day, and 3 days post-infection by i.p. injection with $4 \times 10^7$ PFU of SA 14-14-2. * denotes $P < 0.05$, ** denotes $P < 0.01$, and **** denotes $P < 0.0001$, analyzed by one-way ANOVA with means compared to vehicle-treated control; $n = 4$. **g** WT mice infected with live-attenuated SA 14-14-2 JEV displayed increased MC degranulation in brains, 5 days post-i.p. infection with $4 \times 10^7$ PFU. Magnified image of the MCs in the boxed area is also shown. **h** Human MCs showed MOI-dependent activation to JEV, assessed by β-hexosaminidase assay. ** denotes $P < 0.01$; *** denotes $P < 0.001$ and **** denotes $P < 0.0001$, analyzed by one-way ANOVA with means compared to uninfected control. For **a–c**, a virulent strain, Nakayama, was used. For **d–h**, attenuated strain SA 14-14-2 was used. Scale bar for **a**, **g** is 10 μm, **b** 20 μm, and **d** 5 μm. Data are representative of three independent experiments and error bars represent the SEM. JEV induces MC degranulation in vitro and in vivo

are shown at 5 days post-infection (Fig. 1g, Supplementary Figure 1f). These results demonstrate that multiple strains of JEV activate MCs in vitro and in vivo, including by inducing degranulation of brain-resident MCs during JEV encephalitis.

To support that a similar interaction between JEV and MCs could occur in humans, we investigated whether human MCs are activated upon JEV exposure. Consistent with our data in mice, we observed that cells of the human MC line, ROSA, degranulate significantly in an MOI-dependent manner in response to either live (Fig. 1h) or UV-inactivated JEV (Supplementary Figure 2), establishing that both mouse and human MCs respond to JEV similarly with a virus replication-independent degranulation response.

**MCs increase CNS penetration of JEV**. MCs are innate immune cells and, therefore, we questioned whether MCs promote virus clearance as they do for another closely related *Flavivirus*, DENV[19]. To assess if MCs influence the levels of JEV infection in vivo, we compared peripheral and CNS JEV infection kinetics in MC-competent WT mice to MC-deficient (Kit$^{w-sh/w-sh}$; 'Sash') mice infected with SA 14-14-2 JEV. The Sash mouse is a well-established

MC-deficient mouse model, with a mutation in the regulatory element upstream of *c-Kit* (CD-117), which is essential for MC development[42]. JEV replication was confirmed in both WT and Sash mice by detection of negative-strand viral RNA, which is only produced during virus replication (Supplementary Figure 3a). WT mice cleared infection from the peritoneum, the site of infection, faster than Sash mice starting as early as 6 h after infection (Fig. 2a). However, in spite of enhanced clearance at the site of infection, we did not observe significant differences in viremia by plaque assay (Fig. 2b, Supplementary Figure 3b) or splenic (Fig. 2c) or mesenteric lymph node (Fig. 2d) virus burden between WT and Sash mice. Furthermore, WT and Sash mice showed no differences in serum concentrations of pro-inflammatory cytokines, interferon-γ (IFN-γ), tumor necrosis factor-α (TNF-α), IL-2, IL-4, IL-10, or IL-17 (Fig. 2e, f, Supplementary Figure 3c–f). However, there was an increase in serum IL-6 in the Sash compared to WT mice at the early time point of 6 h (Supplementary Figure 3g). Interestingly, in contrast to the infection kinetics at the site of infection, WT mice showed increased JEV infection levels compared to Sash mice in various parts of the CNS, including cerebral cortex, subcortex, cerebellum, brain stem, and spinal cord (Fig. 2g,

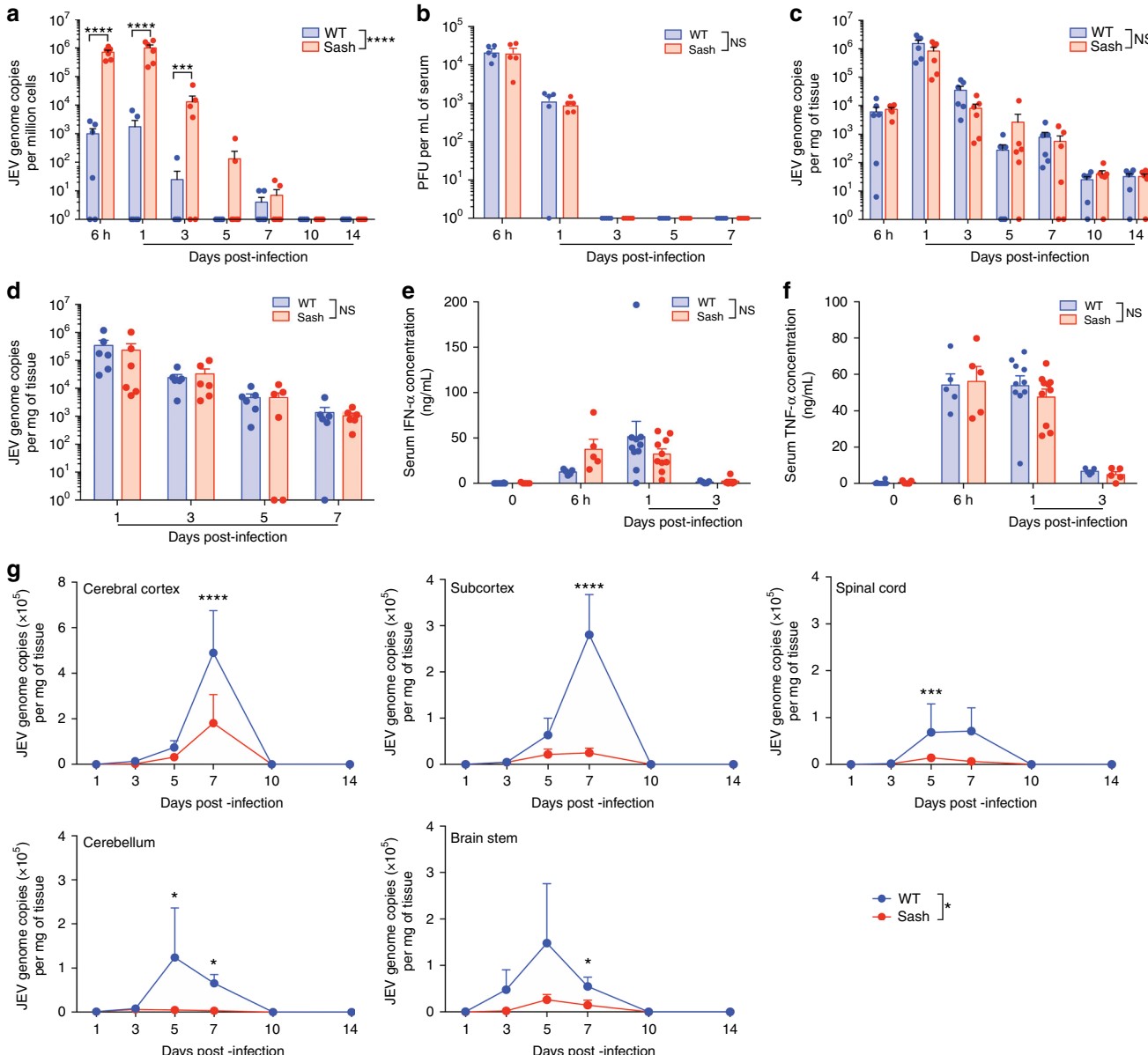

**Fig. 2** Mast cells facilitate JEV clearance from the site of infection but enhance its entry into the brain. Virus burden in **a** peritoneal cells, **b** serum, **c** spleen, and **d** mesenteric lymph nodes was quantified from 6 h to 14 days after i.p. infection with SA 14-14-2 JEV (4 × 10⁷ PFU). Virus genome copies in the peritoneal cells, spleens, and mesenteric lymph nodes were detected by real-time qPCR and virus titers in the sera were detected by plaque assay. WT mice showed increased clearance of JEV in the peritoneal cells but no difference was observed in the serum, spleen, and mesenteric lymph nodes. * denotes $P < 0.05$ as analyzed by two-way ANOVA; $n = 6$. Data are representative of three independent experiments. **e**, **f** Serum of JEV-infected WT and Sash mice had similar levels of multiple cytokines including **e** IFN-γ and **f** TNF-α. NS denotes $P > 0.05$, analyzed by one-way ANOVA; $n = 5$ and results are representative of three independent experiments. **g** Virus quantification in various parts of the brain: cerebral cortex, subcortex, cerebellum, brain stem, and spinal cord from 1 to 14 days after i.p. infection with SA 14-14-2 JEV, as assessed by real-time qPCR. JEV was detected in the brain starting 3 days post-infection and was cleared by 14 days. Virus titers are represented as JEV genome copies per milligram of tissue. * denotes $P < 0.05$, as analyzed by two-way ANOVA; $n = 6$, representative of three independent experiments. Dot-plot presentation of this data is provided in Supplementary Figure 4. Error bars represent the SEM. In spite of similar peripheral JEV titers and inflammatory cytokines, JEV titers were enhanced in multiple brain regions in WT compared to Sash mice

Supplementary Figure 4), throughout the infection time course. The highest infection burden was observed in the cerebral cortex (Fig. 2g, Supplementary Figure 4). JEV was first detected in all parts of the CNS at day 3 and peaked on day 7, with resolution of infection by day 14 post-infection (Fig. 2g, Supplementary Figure 4). In contrast, no difference in JEV infection burden between WT and Sash mice was detected at early time points when the virus was inoculated directly into the brain by intracerebralventricular (i.c.v.) injection, although Sash mice showed delayed clearance of JEV, similar to the peripheral inoculation route (Supplementary Figure 5). These results indicate that the protective effects of MCs were confined to the initial site of inoculation and similar infection and inflammatory profiles were observed on a systemic level in both WT and Sash mice. In contrast, the presence of MCs in vivo led to enhanced JEV infection within the brain after peripheral inoculation.

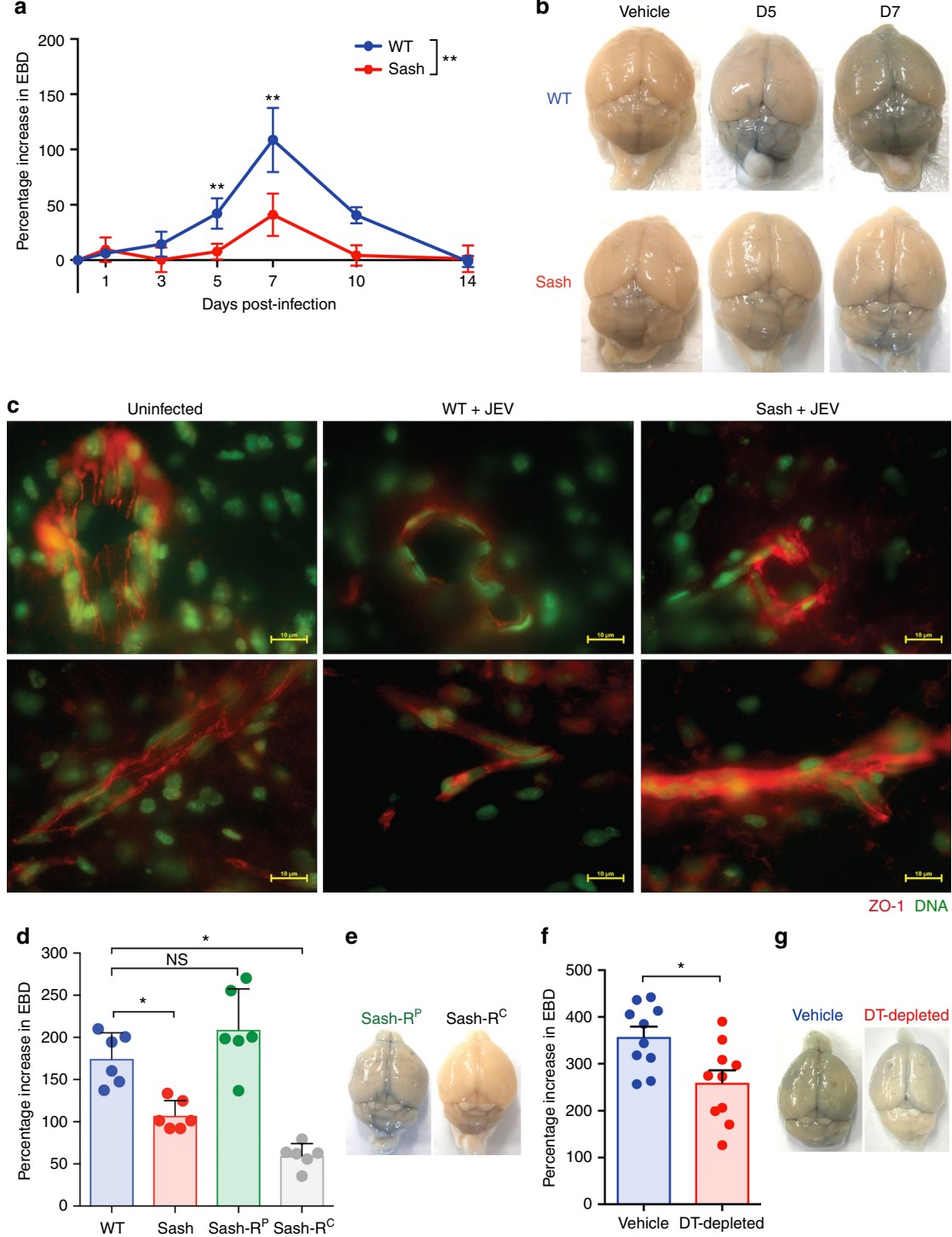

**MCs increase JEV-induced BBB breakdown**. Having observed an increase in CNS infection in MC-sufficient mice, we questioned if JEV-activated MCs affect the integrity of the BBB. For this, we administered Evans blue dye (EBD) via the intra-peritoneal (i.p.) route and assessed its leakage into the brain as a marker for BBB breakdown in both WT and Sash mice, post-infection (Fig. 3a). Quantification of EBD in the brain showed that JEV-infected WT mice experienced worsened BBB compromise, with increased EBD levels in their brains compared to Sash mice (Fig. 3a, Supplementary Figure 6). In contrast to WT mice, which showed extensive leakage of the blue-colored dye, leakage was visibly reduced in the brains of Sash mice (Fig. 3b,

Supplementary Figure 6). Consistent with the increased BBB leakiness, we saw diminished TJ protein ZO-1 staining at the brain endothelial cells in WT-infected mice, 5 days after peripheral JEV infection (Fig. 3c). This reduction was not seen in the MC-deficient Sash mice (Fig. 3c). To fully establish that the phenotype observed in Sash mice is MC-dependent, reconstitution of MCs is usually employed. However, MCs fail to efficiently engraft the brain upon peripheral reconstitution[42]. Therefore, we compared BBB integrity between groups of JEV-infected mice that had been peripherally reconstituted (Sash-R$^P$) and those where BMMCs were injected directly into the CNS (Sash-R$^C$), using an established protocol[43]. A limitation of CNS MC

**Fig. 3** Mast cells increase JEV-induced breakdown of the BBB. **a** Time course of SA 14-14-2 JEV-induced BBB breakdown detected by EBD leakage into the brain from 1 day to 14 days post-infection. Differences in EBD leakage between WT and Sash mice were observed starting 5 days but also at 7 days and 10 days after infection. EBD concentration was normalized to levels detected in vehicle-treated controls. * denotes $P < 0.01$ as analyzed by two-way ANOVA, $n = 6$, representative of three independent experiments. A dot-plot representation of this data is provided in Supplementary Figure 6. **b** Representative images showing extent of EBD leakage into the brain from **a**. **c** Immunofluorescent images comparing ZO-1 staining (red) in brains of uninfected, JEV-infected WT, and JEV-infected Sash mice. Nucleic acid was stained green. Brains were harvested 5 days after i.p. injection of SA 14-14-2 JEV ($4 \times 10^7$ PFU). Brain endothelial cells of JEV-infected WT mice showed decreased ZO-1 levels compared to uninfected controls and JEV-infected Sash mice. Scale bar denotes 10 μm. **d** Comparison of EBD leakage into brains of WT, Sash, Sash reconstituted with BMMCs peripherally (Sash-R$^P$), and Sash reconstituted with BMMCs in the CNS (Sash-R$^C$) at 5 days post-i.p. SA 14-14-2 JEV infection ($4 \times 10^7$ PFU). EBD concentration was normalized to levels detected in vehicle-treated controls and baseline set as 100%. * denotes $P < 0.05$, as determined by one-way ANOVA with Holm-Sidak's post-test; $n = 6$, representative of three independent experiments. **e** Representative images showing the extent of EBD leakage into the brain from **d**. **f** i.c.v Injection of DT (0.5 mg/kg) into *Mcpt5-Cre*; iDTR$^{fl/fl}$ mice depleted ~80% of MCs in the brain (refer to Supplementary Figure 8a, b). Injection of $4 \times 10^7$ PFU of SA 14-14-2 JEV, i.p., in CNS MC-depleted (DT-injected *Mcpt5-Cre*; iDTR$^{fl/fl}$) mice resulted in decreased EBD leakage 5 days post-infection, compared to MC-sufficient control mice. EBD leakage was normalized to uninfected controls. * denotes $P < 0.05$, as analyzed by Student's unpaired two-tailed *t*-test; $n = 10$, representative of two independent experiments. Error bars represent the SEM. **g** Representative images showing extent of EBD leakage into the brain from **f**. Both peripheral and CNS MCs contribute to BBB permeability, yet peripheral MCs are sufficient to induce BBB permeability during JEV infection

reconstitution is lack of efficient reconstitution into the brain tissue and the few reconstituted MCs that were observed did not appear proximal to the BBB (Supplementary Figure 7), where they are located in WT mice (Fig. 1a). In Sash-R$^C$ mice, potentially as a result of inefficient engraftment, we did not see a recapitulation of the vascular leakage phenotype of WT mice (Fig. 3d). This was visualized in the representative image of Sash-R$^C$ brains, which displayed no EBD leakage into the brain (Fig. 3e). In contrast, WT levels of EBD leakage into the brain were recapitulated in Sash-R$^P$ mice 5 days post-infection (Fig. 3d). EBD leakage was readily seen throughout the brain parenchyma of JEV-infected Sash-R$^P$ mice (Fig. 3e). Thus, MCs are critical for JEV-induced vascular leakage at the BBB and our data show that reconstitution of peripheral MCs is sufficient for this breakdown of BBB to occur during JEV infection.

To better understand the contributions of CNS-resident MCs in JEV-induced BBB leakage, an alternative method of MC depletion through direct i.c.v. diphtheria toxin (DT) injection was performed. DT was injected into mice with inducible diphtheria toxin receptor (iDTR) expressed under the MC promoter MCPT5 (Mcpt5-Cre x iDTR$^{fl/fl}$). Inoculation of DT into the brain led to an ~80% reduction in MCs in the brain (Supplementary Figure 8a, b). After i.p. injection with JEV, mice with CNS MC depletion showed decreased BBB breakdown compared to non-depleted controls (Fig. 3f). For normalization, EBD leakage was compared to appropriate controls, including DT-injected MCPT5-Cre mice and PBS-injected Mcpt5-Cre x iDTR$^{fl/fl}$ mice. This reduction can be visualized by representative images of EBD leakage into the brain (Fig. 3g). Thus, MCs are critical for JEV-induced vascular leakage at the BBB and our data show that both peripheral and CNS-resident MCs contribute to BBB breakdown during JEV infection.

**MCs augment BBB leakage, morbidity, and mortality due to JEV.** To assess if MC-induced BBB leakiness can potentiate severe JEV morbidity and mortality, we used the fully virulent Nakayama strain of JEV. To support our data from mice that Nakayama induces MC degranulation (Fig. 1a, b), we confirmed that it also induces degranulation of human MCs much like the live-attenuated strain (Supplementary Figure 9a). Although the Nakayama strain has been used in many mouse studies[39,40], we also confirmed that infectious virus could be detected in the serum in both WT and Sash mice by plaque assay (Supplementary Figure 9b). Subsequently, we measured BBB leakage by quantitating EBD penetration of the BBB at 3 days (Supplementary Figure 9c) and 5 days (Fig. 4a, b) after Nakayama infection. Indeed, we observed an increase in EBD leakage into the brains, suggesting a breakdown of the BBB in WT mice (Fig. 4a, b,

Supplementary Figure 9c). In contrast, EBD leakage was not significantly increased or visually apparent in Sash mice at multiple time points (Fig. 4a, b, Supplementary Figure 9c). Increased BBB permeability coincided with an increase in viral burden in the brain 6 days (Fig. 4c) and 7 days (Supplementary Figure 9d) post-infection in WT mice compared to Sash mice. Thus, MCs promote BBB breakdown and enhanced JEV infection in the brain in a model of lethal JEV encephalitis.

We next examined if MCs could affect morbidity and mortality resulting from the virulent Nakayama strain of JEV. We quantified morbidity by recording the body mass of the animals throughout the course of infection. Significant loss of body mass was observed in MC-competent WT mice (Fig. 4d, Supplementary Figure 10a) beginning around day 4 post-infection. We also quantitated functional neurological deficits through the use of a grip strength meter, which measures the maximal force asserted and indicates the neuromuscular function of the mice[44]. Deficits in grip strength were detected in most of the WT but not in the Sash mice, also beginning around day 4 post-infection (Fig. 4e, Supplementary Figure 10b). Consistent with the increased morbidity, we saw significantly higher mortality of WT mice compared to Sash mice (Fig. 4f). Indeed, 80% of Sash mice survived the challenge, which was lethal for the majority of WT mice by day 7 (Fig. 4f). Therefore, MCs worsen the morbidity and neuro-severity of JEV infection and lead to increased death in a virulent model of JEV infection.

**MC-derived chymase promotes BBB breakdown and CNS infection.** We next investigated the mechanism through which MCs increase BBB permeability during JEV infection. We hypothesized that chymase, one of the MC-specific proteases, could influence the BBB during JEV infection. In support, we noted it to be elevated in the serum during JEV infection (Fig. 1f). To begin to address this hypothesis in vitro, we plated mouse brain endothelial (bEND.3) cells on transwell inserts. Consistent with our in vivo data, supernatants from BMMCs stimulated with JEV led to increased bEND.3 cell monolayer permeability, shown by decreased transendothelial electrical resistance (TEER) across bEND.3 cells (Fig. 5a). This decrease in TEER was not seen in bEND.3 cells exposed to various control conditions including media only, JEV only, or non-stimulated MC supernatant controls (Fig. 5a). When a chymase-specific inhibitor (TY-51469)[45–47] was administered to JEV-stimulated MCs, the TEER was rescued to the levels of control groups (Fig. 5a), including media alone, MC-supernatants alone, and JEV alone groups. Notably, when the MCs were treated with nafamostat mesylate, which is an inhibitor of another MC-protease, tryptase,

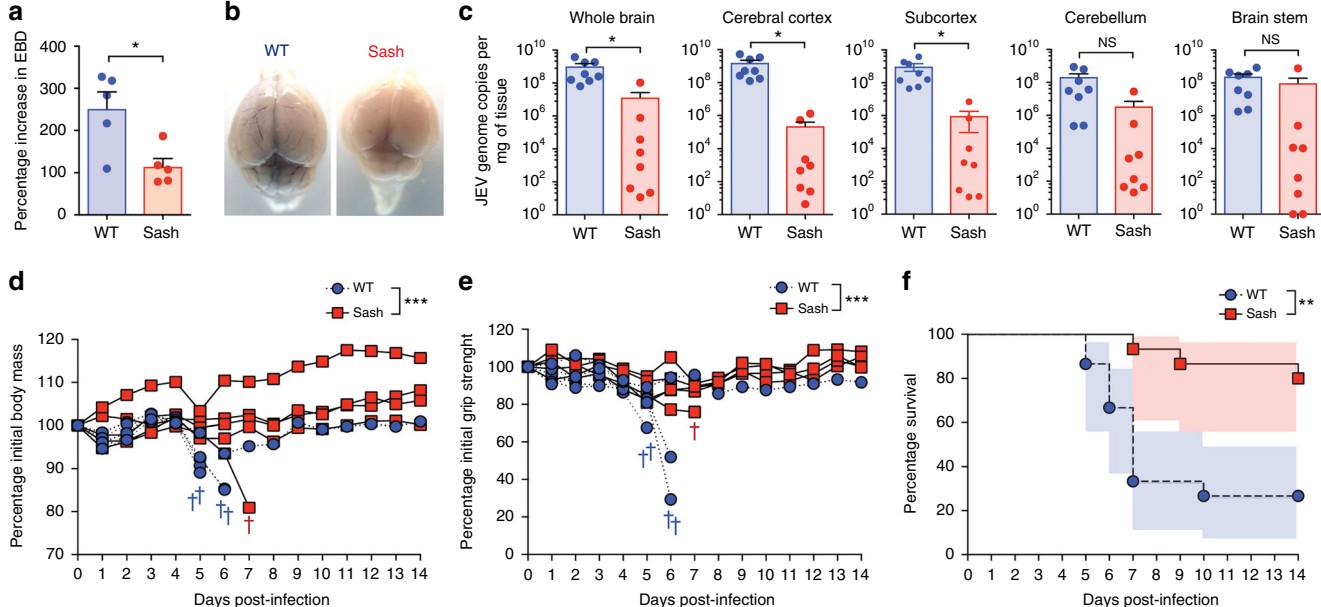

**Fig. 4** Mast cells cause increased vascular leakage and worsened morbidity and mortality during JEV infection. **a** JEV induced increased EBD leakage into the brain in the WT mice but not in the Sash mice at 5 days after infection with JEV Nakayama by i.p. injection of $2 \times 10^7$ PFU. * denotes $P < 0.05$, as analyzed by Student's unpaired two-tailed $t$-test; $n = 5$, representative of three independent experiments. **b** Representative images of EBD leakage into the brain from **a** are presented. **c** WT mice showed increased virus burden in various parts of the brain, including cerebral cortex, subcortex, cerebellum, and brain stem, at 6 days post-Nakayama infection, quantified by real-time qPCR. * denotes $P < 0.01$, as analyzed by two-way ANOVA; $n = 8$, representative of three independent experiments. **d** Body mass and **e** grip strength of mice were measured daily and are presented as a percentage of their initial (day 0) values. WT mice showed increased weight loss and worsened neurologic deficits after Nakayama JEV infection compared to Sash mice. *** denotes $P < 0.001$, as analyzed by two-way ANOVA; $n = 5$, representative of three independent experiments. For **d** and **e**, lines connect points for individual mice and † denotes the last reading before death or humane endpoints were reached. The averages of $n = 15$ mice are included in Supplementary Figure 10. Error bars represent the SEM. **f** Increased mortality and shortened survival were observed in WT mice compared to Sash mice, after i.p. infection with Nakayama JEV. ** denotes $P < 0.01$, as analyzed by log-rank (Mantel–Cox) test, $n = 15$. Shadows represent the 95% confidence intervals. WT mice demonstrated worsened morbidity, neurologic deficits, and mortality compared to Sash mice after peripheral JEV infection

we did not observe the same rescue of TEER breakdown (Fig. 5a). Additionally, we used a secondary measurement of FITC-dextran leakage across bEND.3 monolayers to confirm that chymase inhibition limits endothelial cell permeability, and observed retention of dye by the monolayer in control groups but enhanced leakage of FITC-dextran was observed with exposure to the supernatants of JEV-activated MCs (Fig. 5b). Similarly, chymase but not tryptase inhibition abolished this brain endothelial cell monolayer leakage (Fig. 5b). Finally, transwell experiments were performed with BMMCs produced from transgenic mice deficient in the mouse chymase, MCPT4 (MCPT4-KO). Supernatants of MCPT4-KO BMMCs that were treated with JEV did not induce permeability of bEND.3 cell monolayers, assessed by both TEER and FITC-dextran leakage (Fig. 5c, d). Therefore, we have demonstrated that MC-specific chymase is responsible for JEV-induced BBB compromise, using methods involving both genetic deletion and pharmacological inhibition.

To provide a mechanistic understanding of the components of the BBB affected by MC-released products, we performed western blots to detect BBB TJ proteins under the various conditions of exposure to JEV-elicited MC products or appropriate controls. We observed reduced claudin-5, ZO-1, ZO-2, and occludin in bEND.3 cells exposed to JEV-activated MC supernatant (Fig. 5e, Supplementary Figure 11). Breakdown of these TJ proteins was abolished with administration of the chymase inhibitor TY-51469 at two different doses of treatment (Fig. 5e). These data show that chymase, released from MCs due to JEV activation, increases the permeability of brain endothelial cell monolayers through breakdown of important TJ proteins.

To assess the contributions of chymase to MC-induced BBB leakage and JEV severity and determine therapeutic effects in vivo, we administered TY-51469 to inhibit chymase activity in JEV-infected mice. TY-51469 was confirmed to specifically inhibit chymase enzymatic activity in the serum in vivo (Supplementary Figure 12a), without influencing tryptase activity (Supplementary Figure 12b) or the levels of another MC product released during degranulation, histamine (Supplementary Figure 12c). Interestingly, we observed a reduction in EBD leakage into the brains of TY-51469-treated WT mice, compared to vehicle-treated mice, 5 days after JEV infection (Fig. 5f, g). EBD visualization of BBB permeability in the treated WT mice was similar to the Sash mice (Fig. 5f, g). We also used a secondary method of assessing vascular leakage by staining for IgG, which should be excluded from the brain parenchyma during homeostatic conditions. Consistent with the EBD leakage findings, we again observed that JEV-infected WT mice had increased IgG leakage into the brain parenchyma compared to JEV-infected Sash and uninfected control groups (Fig. 5h). Furthermore, chymase inhibition in JEV-infected WT mice strikingly reduced BBB leakiness (Fig. 5h). Next, we assessed the effect of chymase inhibition on virus penetration into the CNS. Chymase inhibition reduced the viral load in the brain, compared to vehicle-treated WT mice 6 days after JEV infection (Fig. 5i), without affecting the titers of virus in peripheral tissues, including peritoneal cells (Supplementary Figure 13a) and the spleen (Supplementary Figure 13b), which were similar to WT control levels. Finally, to evaluate whether chymase is sufficient as an MC-derived product to induce increased BBB permeability during JEV infection, we injected

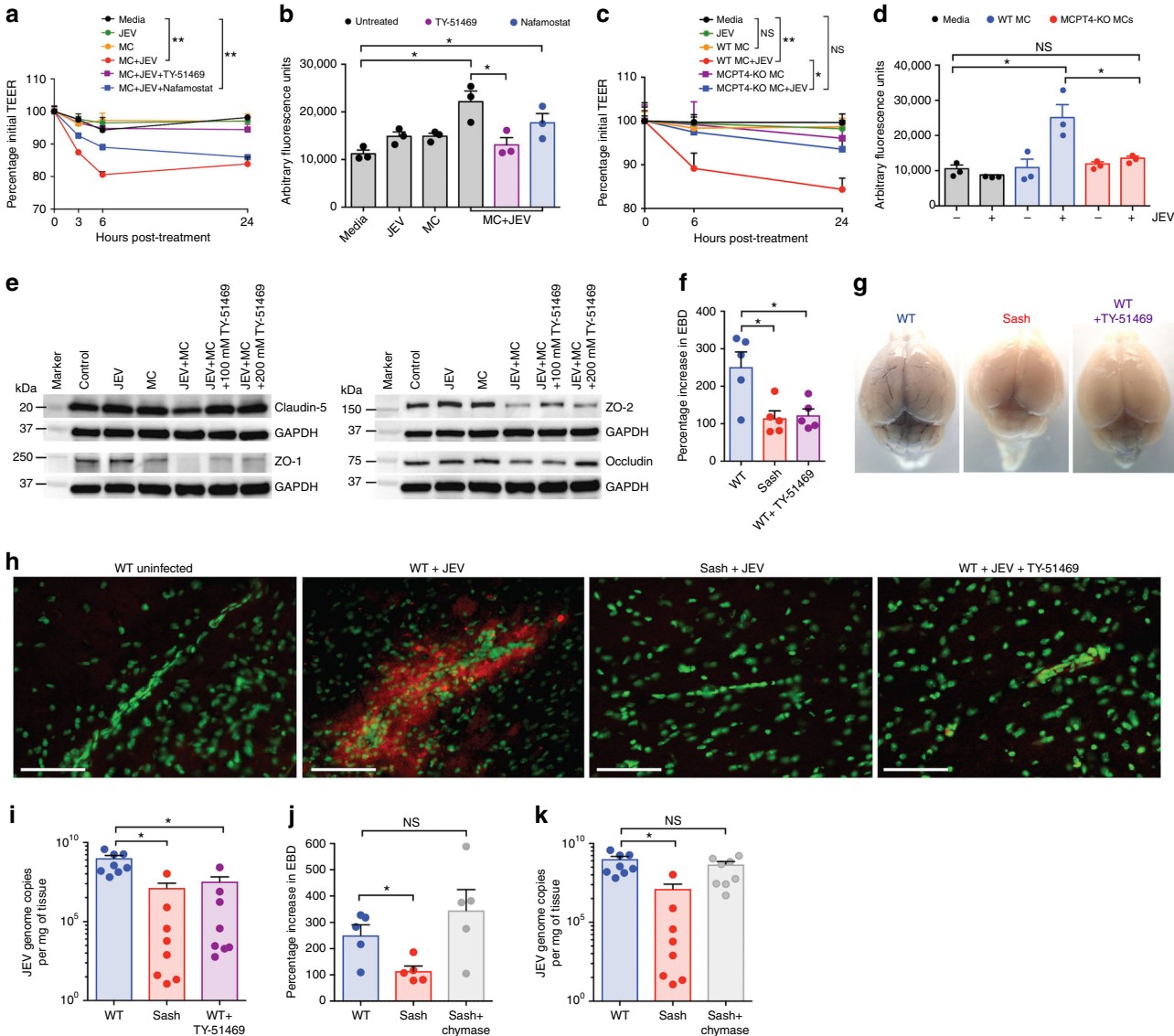

**Fig. 5** Inhibition of chymase abolishes JEV-induced breakdown of the BBB. **a** bEND.3 cell monolayers were treated with media alone or media containing JEV or supernatants of MCs only, JEV-stimulated MCs, or JEV-stimulated MCs treated with either TY-51469 (chymase inhibitor) or nafamostat mesylate (tryptase inhibitor). Supernatants from JEV-stimulated BMMCs reduced the TEER, which was reversed by TY-51469 (100 μM), but not nafamostat mesylate (10 μM); analyzed by two-way ANOVA. **b** JEV-activated BMMC supernatants increased FITC-dextran leakage across transwells 24 h post-exposure. TY-51469, but not nafamostat mesylate, reduced FITC-dextran leakage; $n = 3$. **c** Supernatants from JEV-stimulated WT BMMCs reduced TEER of b.END3 monolayers, but control media, JEV, supernatants from WT or MCPT4-KO BMMCs or JEV-stimulated MCPT4-KO BMMCs did not. **d** JEV-activated WT BMMC supernatants increased FITC-dextran leakage (24 h), but controls or supernatants from JEV-activated MCPT4-KO BMMCs did not; $n = 3$. **e** Claudin-5, ZO-1, ZO-2, and occludin levels in bEND.3 cells exposed to JEV-activated MC supernatants were reduced compared to media, JEV, and unstimulated MC supernatant-treated groups, by western blotting, which was inhibited by TY-51469. GAPDH blotting served as loading controls. Quantification is provided (Supplementary Figure 11). **f** TY-51469 reduced EBD leakage into brains of JEV-infected WT mice, to levels similar to JEV-infected Sash mice, 5 days post-i.p. infection with Nakayama ($2 \times 10^7$ PFU); $n = 5$. **g** Representative images from **f**. **h** IgG (red) of brain sections of JEV-infected WT, Sash, and TY-51469-treated WT mice, 5 days post-i.p. infection with Nakayama. During JEV infection, IgG was detected in the brain parenchyma of WT mice, but not Sash or TY-51469-treated WT mice. Scale bar = 50 μm. **i** TY-51469 reduced the JEV titers in brains, 6 days post-i.p. Nakayama infection, compared to vehicle-treated WT mice, to levels similar to Sash mice. Chymase injection (30 ng, i.p.) of Sash mice led to increased **j** EBD leakage ($n = 5$) and **k** brain JEV titers, to levels similar to WT JEV-infected mice (5 days post-infection); $n = 8$; representative of two experiments. Error bars represent the SEM. Unless indicated, all data are representative of three independent experiments, were analyzed by one-way ANOVA with Holm-Sidak's multiple comparison test; * denotes $P < 0.05$ and ** denotes $P < 0.01$. Chymase inhibition limits JEV penetration of the brain by preventing BBB leakage

purified enzymatically active chymase into JEV-infected Sash mice at days 1 and 3 post-infection. We observed an increase in EBD leakage into the brains (Fig. 5j) and enhanced titers of JEV in the CNS of chymase-injected Sash mice compared to vehicle-treated group (Fig. 5k), demonstrating that chymase alone is

sufficient to enhance BBB permeability and JEV penetration into the brain.

We next questioned whether therapeutic targeting of chymase in vivo could improve measures of severe disease and neurological dysfunction in mice. In TY-51469-treated compared

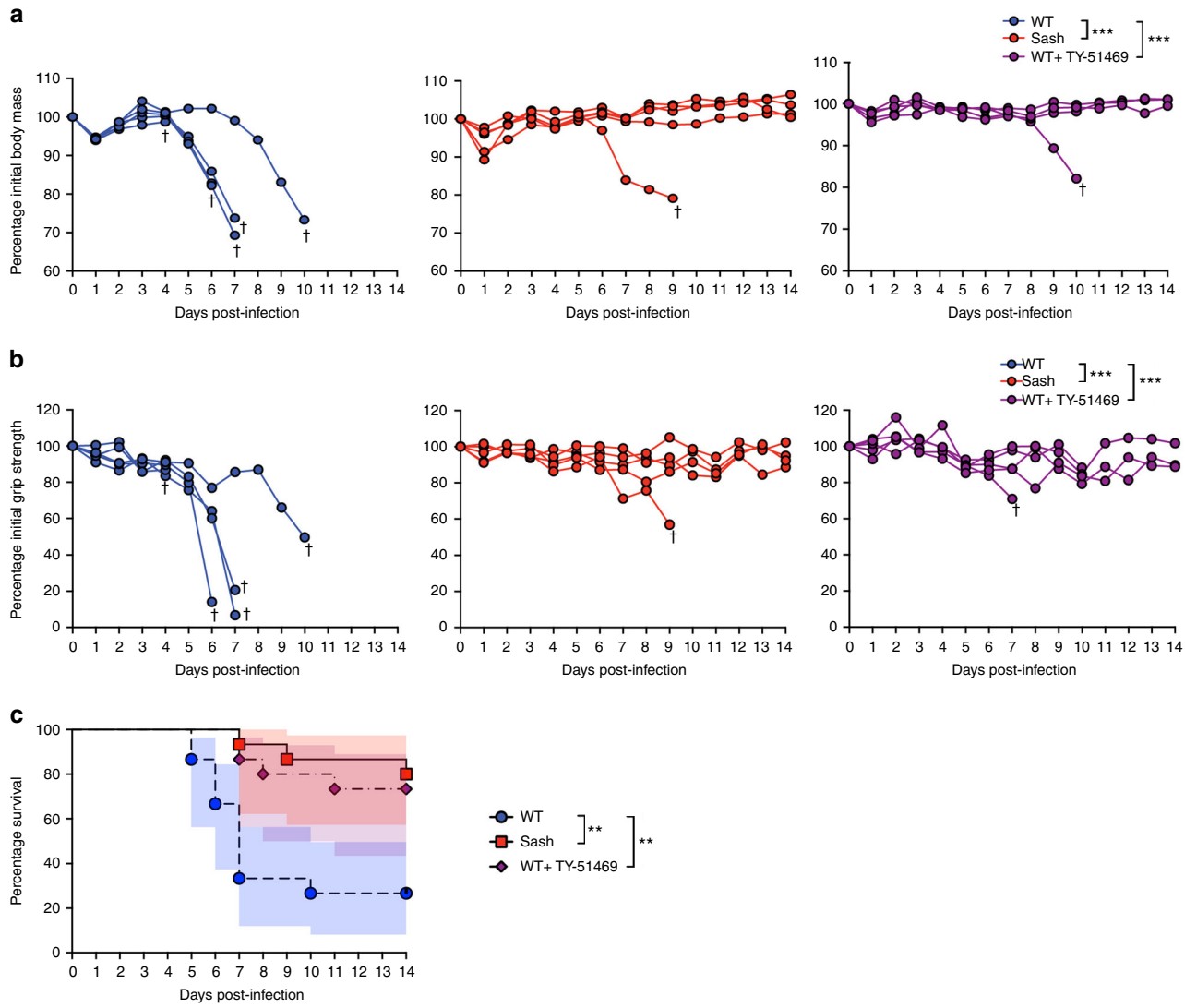

**Fig. 6** Chymase inhibition is therapeutic during lethal JEV infection. **a** Body mass and **b** grip strength of mice were measured daily and are shown as a percentage of pre-infection measurements from day 0. Chymase inhibition with TY-51469 maintained body weight and reduced grip strength deficits in WT mice after Nakayama JEV infection to levels similar to Sash. *** denotes $P < 0.001$, as analyzed by two-way ANOVA, $n = 5$, representative of three independent experiments (all data from $n = 15$ mice are included in Supplementary Figure 10). Error bars represent the SEM. **c** Survival curve of WT, Sash, and WT mice treated with TY-51469 after i.p. infection with Nakayama JEV is presented. TY-51469 administration reduced mortality and prolonged survival in WT mice. * denotes $P < 0.05$, ** denotes $P < 0.01$ as analyzed by log-rank (Mantel–Cox) test; $n = 15$. Shadows represent the 95% confidence intervals. Chymase inhibition is therapeutic in reducing morbidity, neurological deficits, and mortality during lethal JEV infection

to vehicle-treated JEV-infected WT mice, we observed maintained body mass (Fig. 6a, Supplementary Figure 10a) and significant improvement in grip strength measurements (Fig. 6b, Supplementary Figure 10b). Moreover, JEV-infected mice treated with TY-51469 showed significantly improved survival from lethal JEV infection compared to WT mice, similar to Sash mice (Fig. 6c). These findings show that MC-derived chymase increases JEV-induced permeability of the BBB, augments viral infection in the CNS, and ultimately worsens morbidity and mortality of the infected host. Furthermore, chymase is a potential therapeutic target to limit JEV infection, neurological deficits, and death.

## Discussion

JEV neurovirulence and neuropenetration cause encephalitis, a signature clinical pathological outcome associated with human JEV infection. Our results provide a previously unrecognized

mechanism of JEV penetration of the BBB that is dependent on MCs and, specifically, the MC-specific vasoactive protease chymase. MCs have not previously been implicated in worsening Japanese encephalitis. Yet, we observed that JEV infection is worsened in the brain due to the activation and degranulation of MCs, involving the release of granule-associated proteases, including chymase. This contrasts the well-established role of MCs as innate immune sentinels that coordinate clearance of invading pathogens[15], although we did observe that MCs promoted enhanced clearance of JEV from the site of inoculation. In spite of their protective effect in the periphery, peripheral MCs were also a source of vasoactive products, including chymase, which is sufficient to initiate BBB permeability. In turn, BBB permeability appears to have provided enhanced access of JEV to the CNS. MCs and, specifically, MC-derived chymase increased leakage at the BBB due to cleavage of TJs, leading to increased infection within various CNS compartments (including the

cerebral cortex and other regions of the brain), neurological deficits, and reduced survival upon challenge with a fully virulent strain of JEV. Furthermore, we showed that chymase is a potential therapeutic target to prevent JEV CNS infection and associated morbidity and mortality since these negative outcomes are reversed by treatment with a chymase-specific inhibitor.

Penetration of the BBB is a key component of JEV pathogenesis, leading to brain infection and encephalitis. Multiple mechanisms through which this could occur have been postulated, including direct infection of brain microvascular endothelial cells[48], trafficking of infected cells into the brain in a Trojan horse method[49], and peri-cellular transfer into the brain following BBB breakdown[7,12]. Cell culture-based experiments have suggested that resident supporting cells of the neurovascular unit, including pericytes[50] and astrocytes[51], could amplify BBB permeability, but it is unclear how these cell types would be substantially infected in vivo prior to breaching of the BBB. Here, we have identified MC activation as an initiating step of BBB breakdown in vivo during JEV infection. This breakdown of the BBB subsequently led to increased infection, morbidity, and mortality.

Although quintessential pathogen surveillance cells, MCs, are also located in the brain where infections are rare. However, to-date, it has been unclear how MCs respond to viral encephalitis when it does occur. Our results establish that CNS-resident MCs are able to degranulate significantly during infection. Yet, this MC degranulation does not depend on virus replication within the MCs, as UV-inactivated, replication-deficient JEV can still activate MCs. While we primarily describe here the influence that MC degranulation has on promoting BBB leakage and JEV infection in the CNS, which would contribute directly to the neurological deficits observed, it is also possible that the resulting inflammation due to activating MCs in the brain is a contributing factor to neurological deficits and morbidity. The contributions of MCs to JEV-induced BBB leakage and pathophysiology are also consistent with the evidence that they promote vascular leakage and BBB compromise during sterile inflammatory events. For example, acute stress has been shown to increase BBB permeability in an MC-dependent manner[26]. Pharmacological stabilization of MCs suggested that they contribute to worsened disease severity in ischemic stroke[25,28,31] and hemorrhagic stroke induced by ischemia reperfusion[27]. Our observations that MCs promote BBB breakdown and lead to increased viral titers in the CNS support that MC activation is an essential event initiating BBB breakdown during viral encephalitis, rather than a byproduct of infection.

Since MCs of the periphery and CNS may contribute differently during viral encephalitis, we aimed to delineate their individual contributions to host defense or pathogenesis. To study the effect of CNS-resident MCs, we first attempted reconstitution of brain MCs through i.c.v. injection. In these mice, we failed to see a recapitulation of the WT leaky BBB phenotype with JEV infection. However, we noted inefficient engraftment after i.c.v. injection and the MCs observed in the brain were not perivascular but, rather, in the meninges and, occasionally, in the brain parenchyma. Thus, to better assess the function of CNS-resident MCs, we used an alternative method for MC ablation, a mouse model with an inducible-DT receptor knocked-in under the control of the *Mcpt5* promoter of MCs. We optimized a novel method of DT administration into the brain, which led to specific ablation of CNS-resident MCs including perivascular MCs, while the peripheral MCs remained intact. We observed a reduction in BBB compromise during JEV infection in this system when CNS-resident MCs were selectively ablated. This suggests, like peripheral MCs regulate vascular permeability, CNS-resident MCs also regulate BBB permeability and can break down BBB TJs. We

expect that following initiation of BBB permeability by peripheral MC activation, CNS-resident MCs can further amplify the breakdown of the BBB during infection. This optimized CNS ablation model can be used in future studies to delineate functional roles of peripheral vs. CNS-resident MCs in other diseases and pathologies.

We also showed that chymase, a chymotrypsin-like serine protease produced specifically by MCs, is the primary MC-derived mediator promoting BBB breakdown during JEV infection. Both in an in vitro trans-well assay and in vivo in mice using multiple strains of JEV, MC-induced BBB leakiness was reversed with treatment using a chymase-specific inhibitor, indicating a novel therapeutic target of disease. Mechanistically, we observed that chymase breaks down important TJ proteins, including ZO-1, ZO-2, claudin-5, and occludin. In vitro studies have suggested that occludin and claudin have sites that can be directly cleaved by chymase[52]. Moreover, chymase can activate other potent proteases such as MMP2 and MMP9 into their enzymatically active forms[36], which may be another mechanism of chymase's function in this system. Activated MMPs can increase endothelial cell permeability[53] and promote BBB breakdown[54]. Interestingly, MMP9 has been shown to enhance penetration of West Nile virus, which is closely related to JEV, into the brain[55]. For DENV, MC-derived cytokines such as TNF and IFNs can increase the permeability of endothelial monolayers[18,56,57]. Importantly, we did not observe increased cytokines known to induce BBB leakiness, such as IFN-γ and TNF-α, in MC-sufficient mice compared to Sash mice, further supporting that a MC-derived product is responsible for the initial breakdown of the BBB. The small molecule inhibitor of chymase used here, TY-51469, has been tested as a treatment for other sterile inflammatory diseases such as myocardial infarction[46], inflammatory bowel disease[58], non-alcoholic steatohepatitis[47], and pulmonary fibrosis[45] in preclinical models. Thus far, it has been shown to be safe and efficacious in animal models of these diseases.

We have previously reported that chymase is released by MCs in response to another related virus, DENV, and can be a biomarker of DENV disease severity in animals and humans[20,59]. However, neurological complications of DENV are more rare and not a consistent feature of disease. This could potentially be attributable to differences in tropism for the two viruses for the cells in the brain. For example, GRP78, which is expressed on neurons, is a receptor for JEV that is critical for the cell entry stage of the JEV life cycle[60]; however, for DENV, GRP78 does not appear to be an obligate entry receptor, although its expression can enhance viral replication[61]. This suggests that, even within the *Flavivirus* genus, there are differences in the capacity of unique viruses to utilize receptors to infect and replicate in the CNS after gaining access to it.

To induce encephalitis, JEV needs to not only enter the brain (neuropenetrance), but also establish infection in neural tissue (neurotropism) and cause neuronal dysfunction (neurovirulence). This point is supported by our comparison of the infection kinetics and morbidity/morality between a virulent (Nakayama) and a live-attenuated vaccine (SA 14-14-2) strain of JEV. We showed that both strains caused BBB compromise in mice, in an MC-dependent manner. Further, they both infect primarily neurons, consistent with previous reports[7]. However, the Nakayama JEV produced significantly higher brain infection, elicited greater neurologic deficits, and caused more mortality as compared to the SA 14-14-2 live-attenuated strain. Indeed, consistent with previous studies[62,63], even when the SA 14-14-2 virus is inoculated directly into the mouse brain, it failed to cause mortality, indicating intrinsic reductions in neurovirulence through mechanisms independent of neuropenetrance. This illustrates that Japanese encephalitis development is a multi-

faceted process and multiple steps can be targeted for therapeutic development.

Here, we have shown that MC activation by JEV reduces BBB integrity during CNS infection but it remains unknown whether MCs regulate BBB permeability or influence infection by other neurotropic viruses. Genetic models and pharmacological targeting support that MCs and the MC-derived protease, chymase, assert a functionally significant role in the compromise of the tightly regulated BBB during JEV infection and facilitate penetration of JEV into the CNS. Importantly, our in vivo preclinical data suggest that therapeutically inhibiting chymase can reduce JEV penetration of the BBB and reduce the signs and mortality associated with Japanese encephalitis.

## Methods

**Animal studies and infections.** Control mice on a C57BL/6 background were purchased from InVivos (Singapore). MC-deficient mice ($Kit^{w-sh/w-sh}$; 'Sash') and iDTR mice were originally purchased from Jackson Laboratories and bred in-house. Mcpt5-Cre mice were provided by Axel Roers, Dresden University, Dresden, Germany. Male mice (6-weeks old) were used for all experiments and littermates randomly assigned to experimental groups. Peripheral JEV infection was performed by i.p. injection of $4 \times 10^7$ PFU for SA 14-14-2 and $2 \times 10^7$ PFU for Nakayama JEV. CNS JEV infection was performed by i.c.v. injection of $1 \times 10^4$ PFU of SA 14-14-2 and 100 PFU of Nakayma JEV. All animal experiments were performed according to protocols and ethical guidelines approved by the SingHealth Institution Animal Care and Use Committee (ref. no.: 2015/SHS/1077).

**In vivo MC reconstitution or depletion.** To generate BMMCs for both in vitro experiments and in vivo reconstitution studies, bone marrow was flushed from mouse femurs and cultured in RPMI medium containing 10% FBS, penicillin and streptomycin, HEPES, 1% supernatant from Cho-KL cells containing stem cell factor (produced in-house), and recombinant IL-3 (5 ng/mL, R&D Systems, Minneapolis, MN; #203-IL-010). MCs were verified to be >95% pure by toluidine blue staining (Sigma-Aldrich, St. Louis, MO) prior to use. Peripheral reconstitution of Sash mice was performed by injecting $1 \times 10^7$ BMMCs i.v., followed by a 6-week period to allow complete engraftment, as previously described[42,64]. CNS reconstitution of Sash mice was performed by injecting $1 \times 10^6$ MCs i.c.v., followed by a 16-week period for engraftment in the brain[43]. For depletion of CNS-resident MCs in the *MCPT5-CRE*; iDTR$^{fl/fl}$ mice, DT (dose 0.5 µg/kg) was injected i.c.v. into the brain at days 0 and 4, with experiments performed on day 6. This dose was found to be optimal because higher doses of DT were observed to be toxic. For administration of chymase, 30 ng of purified chymase (Sigma-Aldrich, St. Louis, MO; #97501-92-3) is injected i.p. at days 1 and 3 post-infection of Sash mice.

**Chymase inhibitor treatment.** For drug treatment, chymase inhibiting drug (TY-51469, synthesized by the Duke Small Molecule Synthesis Facility, Durham, NC) was given by i.p. injection (0.5 µg/g body weight) just before infection and daily thereafter.

**Neurological deficit (grip strength) measurement.** To assess neurological deficits, a grip strength meter (TSE Systems, Bad Homburg, Germany) was utilized to assess four-limb strength. Each mouse was allowed to grab with four limbs onto a mesh connected to a force gauge. As a mouse grasped the mesh, an investigator pulled back its tail, and the force of grip was measured and recorded on a force transducer. The peak force was measured when the mouse released its grasp. The force transducer was reset to zero before every trial. All grip strength tests were measured in triplicates, with 5 min rest between trials.

**Cell lines and virus strains.** Human ROSA MCs[65] were obtained from Michel Arock and cultured at 37 °C in RPMI medium supplemented with penicillin/streptomycin, fetal bovine serum (FBS), and 1% supernatant from Cho-KL cells, which contains stem cell factor. Mouse brain endothelial bEND.3 cells were obtained from ATCC (CRL-2299) and cultured in DMEM medium supplemented with penicillin/streptomycin and FBS. Sub-culturing was performed using trypsin-EDTA (Thermo Fisher, Waltham, MA; #25200056). The low-passage SA 14-14-2 and Nakayama strains of JEV used for infections were provided by Eng Eong Ooi. Virus strains were propagated in *Aedes albopictus* C6/36 mosquito cells (CRL-1660; ATCC), maintained at 28 °C in RPMI medium 1640 with 25 mM HEPES, and titered using standard methods[66]. All cell lines were verified to be mycoplasma-negative using a kit (C7028, Thermo Fisher Scientific).

**Measurement of MC degranulation.** MC degranulation was assessed in vitro by a standard β-hexosaminidase assay. About $5 \times 10^5$ human ROSA cells and mouse BMMCs were exposed to JEV at an MOI of 1, 3, 5, and 10 for 1 h, after

which β-hexosaminidase levels were measured in the supernatant as well as the cell lysate solubilized in 0.1% Triton X-100, according to published protocols[19]. β-hexosaminidase levels were detected by p-nitrophenyl-N-acetyl-β-D-glucosaminide in 0.1 M sodium citrate (pH 4.5) for 1 h at 37 °C, after which 0.1 M carbonate buffer (pH 10) was added to stop the reaction. End-product 4-p-nitrophenol was detected at absorbance of 405 nm. Percentage degranulation was calculated by dividing the absorbance in supernatant with the sum of absorbance in both supernatant and cell lysate. MC degranulation in vivo was assessed in the mice by detection of MC-specific product, chymase, in the serum. After i.p. injection of JEV, blood samples were collected via the maxillary vein at various times post-infection and serum isolated to measure mouse chymase (MCPT1) concentrations using an ELISA kit (Thermo Fisher, Waltham, MA; #BMS6005).

**Quantification of BBB breakdown.** EBD was used to assess BBB permeability as described by published protocols[67,68]. For this, 800 µL of 1% (w/v) of EBD (Sigma-Aldrich, St. Louis, MO; #314-13-6) was injected i.p. in each mouse. After 1 h, the mouse was euthanized and perfused thoroughly with 60 mL cold PBS through intra-cardiac puncture. The mouse brain was recovered immediately, weighed, and the extent of EBD leakage captured with a high-resolution digital camera. To quantify EBD leakage, the brain was homogenized in 0.5 mL of PBS with homogenizing beads (Glen Mills Inc., Clifton, NJ; #7305-000002) for 15 min, and proteins precipitated by 0.5 mL of 100% trichloroacetic acid (Sigma-Aldrich, St. Louis, MO; #T6399). The mixture was vortexed for 2 min and cooled for 30 min at 4 °C. After centrifugation at 4000×g for 30 min, 150 µL of the supernatant was collected for measurement of optical density by a spectrophotometer (Tecan Infinite M200) at 620 nm. EBD levels were quantified based on a standard curve and expressed as micrograms of the dye per milligram of brain tissue.

**Quantification of JEV virus titer.** RNA was extracted from the mouse tissues including brain, spinal cord, and spleen, using TRIzol (Thermo Fisher, Waltham, MA) and from mesenteric lymph nodes and peritoneal cells using RNeasy mini kit (Qiagen, Venlo, Netherlands; #74104). cDNA was made using iScript cDNA Synthesis Kit (Bio-rad) and subsequently quantified using SybrGreen Real Time PCR Master Mix (Thermo Fisher, Waltham, MA). Previously published primers targeting the 5′ UTR region of JEV were used[69] and are included in Supplementary Table 1. The reaction conditions used for PCR were 95 °C for 15 min, 40 cycles of 95 °C for 15 s, 55 °C for 30 s, and 72 °C for 10 s followed by a melting curve step. RNA was extracted, cDNA was synthesized, and qPCR reactions were plated with RNAse-free water. RNA isolated from uninfected mouse tissues was used as negative control. Correct amplification was verified by viewing melt curves and PCR products run on agarose gels to visualize the correct band size for the amplification product. Negative control (uninfected) tissues were run with each assay to ensure specific amplification without contamination. Sequencing was also used to verify correct amplification during the optimization of the assay. pUC19 plasmids were inserted with appropriate JEV sequences to construct standard curves for quantifying the virus genome copies.

**Negative-strand PCR for JEV.** Using the RNA isolated from tissues, as described above, cDNA was made using the iScript cDNA Synthesis Kit (Bio-rad) using RNAse-free water (Qiagen) under sterile conditions to prevent contamination. A previously published primer (Supplementary Table 1) was used to generate cDNA of the negative-strand. The JEV NS5 region of the negative-strand was subsequently amplified using Platinum™ Hot Start PCR Master Mix (Thermo Fisher, Waltham, MA) using previously published primers[70] that are also provided in Supplementary Table 1. The reaction conditions used for PCR were 95 °C for 15 min, 40 cycles of 95 °C for 15 s, 55 °C for 30 s, and 72 °C for 10 s. The PCR products were subsequently run on a 1% agarose gel to visualize the product bands specific to the primers. A 100-bp ladder (Promega, Madison, WI) was utilized to identify the 250bp PCR product. An uncut gel image is provided in Supplementary Data 1.

**Microscopy.** To examine the activation status of MCs, JEV infection, and TJ protein integrity of the brain, 15-µm frozen brain sections were prepared. For confocal microscopy, the sections were fixed with paraformaldehyde, permeabilized using 0.3% Triton X-100 in PBS for 30 min at room temperature and subsequently blocked for 1 h at room temperature in blocking buffer (3% BSA in PBS). Tissue sections (or cells) were stained using the following primary antibodies overnight: rabbit antibody against TJ protein ZO-1 (Thermo Fisher, Waltham, MA; #40-2200, 1:500) or JEV NS3 (GeneTex, Irvine, CA; #GTX125868, 1:250), rat antibody against mouse CD31 for brain endothelial cells (BD Biosciences, Singapore; #550274, 1:250), goat antibody against mouse IgG (H + L chain) conjugated to AlexaFlour555 (Invitrogen, Oregon, USA #A21422, 1:500) for IgG leakage and the MC-specific probe, avidin conjugated to FITC (Thermo Fisher, Waltham, MA; #434411, 1:500) or TRITC (Sigma-Aldrich, St. Louis, MO; #A7169, 1:500). After washing, tissue sections were subsequently stained with the following secondary antibodies: anti-rabbit Cy3 (Jackson ImmunoResearch, West Grove, PA; #711-165-152, 1:1000) and anti-rat AlexaFlour647 (Jackson ImmunoResearch, West Grove, PA; #712-606-150, 1:1000) for 1 h at room temperature. Samples were washed with PBS and mounted with ProLong™

Gold antifade reagent with DAPI (Thermo Fisher, Waltham, MA; #P36935). Confocal images of stained brain sections and cells were obtained with a three-laser Nikon confocal laser-scanning instrument with a channel series approach to diminish spectral overlap. For toluidine blue staining of mouse tissues, sectioned brain tissues were fixed with Carnoy's fixative (60% ethanol, 30% chloroform, and 10% glacial acetic acid) before staining with 0.1% solution of toluidine blue stain (pH 2.3). For H&E staining of the mouse tissues, sections were stained with hematoxylin (Sigma-Aldrich, St. Louis, MO; #MHS16) for 5 min followed by rinsing in tap water and distilled water before staining with Eosin-Y (Sigma-Aldrich, St. Louis, MO; #230251) for 1 min. Sections were rinsed in distilled water and dehydrated through a series of alcohol gradients (50, 70, 95, and 100% ethanol), air dried and mounted using a permount solution (Thermo Fischer, Waltham, MA; #SP-15–500) for visualization under a light microscope. To visualize the activation status of BMMCs, the cells were exposed to JEV at an MOI of 1 for 1 h in cell culture conditions, followed by cytospinning onto slides. Images were prepared for publication with ImageJ software.

**Transendothelial permeability assays**. Mouse brain endothelial bEND.3 cells ($5 \times 10^5$ cells in 500 μL DMEM media, passage number 4 or 5) were grown for 1 day on 3 μM transwell inserts (BD Biosciences; Corning, #353492) until confluency and tight junctions were well-established. Media were removed from the transwells and the supernatants harvested from various experimental conditions of BMMCs were added. Barrier function of the endothelial monolayers was measured as TEER using a Millicell ERS Ohmmeter with the probe included by the manufacturer (Millipore, Burlington, MA; MERS00002). TEER readings were taken at baseline, 3h, 6h, and 24 h after introduction of stimulus. For transwell assays using BMMCs, an MOI of 1 was used. For some groups, chymase-specific inhibitor (TY-51469; 100 μM) and tryptase-specific inhibitor (nafamostat mesylate; 10 μM, Sigma-Aldrich, St. Louis, MO; #82956-11-4) were incubated with BMMCs during JEV activation. At 24 h after bEND.3 cell stimulation with the various experimental supernatants, FITC-dextran was added to the upper compartments at 0.5 mg/mL for 1 h, then the medium was removed from the lower compartments for fluorescence measurement. Fluorescence was measured using a plate reader (Tecan Infinite M200; excitation wavelength of 492 nm and emission wavelength of 520 nm).

**Western blot analysis of TJ proteins**. The cells were washed with ice-cold PBS and lysed using a cell scraper with lysis buffer (1% NP40 in Tris-HCl pH8.0, 150 nM NaCl, 1 mM EDTA, 1 mM EGTA, and protease inhibitors) on ice. After centrifugation, the supernatants were harvested and a BCA assay performed to assess for protein concentration. The protein lysates were added to 2× laemmli buffer (Bio-rad, Hercules, CA; #1610737) such that equal proteins would be loaded for each experimental condition. Proteins were then separated by SDS-PAGE and electrophoretically transferred to methanol-activated polyvinylidene difluoride membranes. The membranes were blocked for 1 h with dry skim milk dissolved in 0.1% Tween-20 in TBS. After blocking, the membranes were incubated with the following primary antibodies at 4 °C for overnight: rabbit anti-claudin-5 (Thermo Fisher #34–1600; 1:250), rabbit anti-ZO-1 (Thermo Fisher #40–2200; 1:250), rabbit anti-ZO-2 (Thermo Fisher #71–1400; 1:250), rabbit anti-occludin (Thermo Fisher #71–1500; 1:250), or rabbit anti-GAPDH (Cell Signaling #2118; 1:1000). After washing, the membranes were incubated with HRP-conjugated anti-rabbit IgG (Jackson Immunology, #111–035–144; 1:2500) for 1 h at room temperature. The membranes were washed again and the protein bands visualized with ECL buffer (GE Life Sciences, Pittsburgh, PA; #45-000-875). The membranes were visualized by the GelDoc™ imaging system (Bio-rad, Hercules, CA). The original, unprocessed blot images are provided in Supplementary Data 1.

**Cytokine profiling**. Levels of cytokines, including IFNγ, TNFα, IL-2, IL-4, IL-6, IL-10, and IL-17 were assessed by the mouse Th1/Th2/Th17 cytokine kit (BD Biosciences, Singapore; #560485). Serum was collected from mice and used for the assay according to manufacturer's instructions. Cytokine levels were analyzed by flow cytometry (LSRFortessa, BD Biosciences, Singapore).

**Statistical analysis**. Prism 5 and Excel were used to determine statistical significance. For direct comparisons of infected and control samples, Student's unpaired two-tailed $t$-test was used. For comparisons of multiple groups, one- or two-way ANOVAs were performed, as appropriate, with Bonferroni's post-test to determine statistical significance. For comparison of survival between treated and control groups, log-rank (Mantel–Cox) test was used. Statistical details of each experiment can be found in the figure legends. The data were considered significant at $P \leq 0.05$.

**Reporting Summary**. Further information on experimental design is available in the Nature Research Reporting Summary linked to this Article.

## Data availability

The data needed to evaluate the conclusions of this paper are present in the paper and/or the Supplementary Materials. Additional data related to this paper may be requested from the authors.

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

## Acknowledgements

The authors thank Michel Arock (Pitié-Salpêtrière Hospital, Paris, France) for permission to use the ROSA MC line, Ooi Eng Eong (Duke-NUS Medical School, Singapore) for providing Nakayama and SA 14-14-2 strains of JEV, Axel Roers (Dresden University, Dresden, Germany) for providing *MCPT5-cre* mice and Magnus Åbrink (Swedish University of Agricultural Sciences, Uppsala, Sweden) for *MCPT4−/−* bone marrow. The authors also thank Duke-NUS Medical School ABSL3 facility manager Benson Ng and staff Velraj S/O Sivalingam for assistance with experiments and BSL3 operations. Youngjoo Choi is thanked for advice on flow cytometry analysis of cytokine data. This work was supported by Duke-NUS start-up funds.

## Author contributions

J.T.H., A.P.S.R., and A.L.S. designed the experiments. J.T.H., A.P.S.R., and G.S. performed the experiments. J.T.H. and A.L.S. analyzed the data and wrote the manuscript. All authors read and approved the final manuscript.

## Additional information

**Competing interests:** The authors declare no competing interests.

