## [Peer Review File · Nature Communications]

Point-by-point response to reviewers' comments:

We thank the reviewers for their supportive comments and constructive suggestions.

Reviewer 1:

1. The principal problem is the issue of animal numbers used for survival experiments (Figures 4 and 6). Animal number of only five per group is simply not enough to draw any conclusion from these experiments. None of the mice experiments were repeated for reproducibility of the results.

Our weight loss, grip strength and survival experiments (**Figures 4 and 6**) have all been performed as 3 experimental repeats for a total n=15. We prefer our presentation of the data showing one experimental repeat because it allows visualization of the results of individual mice; however, we have now included supplemental figures that provide the averages of all 15 mice from 3 experimental repeats (**Fig. S10**), which correspond to **Figures 4d-e, 6a-b**. For **Figure 4f** showing survival, the graph has been replaced with a one containing n=15 (added from 3 experimental repeats involving n=5 per group). Numbers of experimental replicates have been added to the appropriate places in the figure legends.

2. Conclusion that “MCs limit peripheral JEV infection while increasing JEV penetration into the CNS” is not supported by the data presented. Only peritoneal cells and spleen were analyzed for peripheral virus replication. Blood/Serum and lymph nodes, which I think are most important, were not analyzed in this study. Other peripheral organs such as kidneys or liver were also not analyzed. It is known that high peripheral viremia during WNV and JEV infections results in high brain viral load, thereby leading to severe encephalitis.

We have now added the requested data. First, we show that MCs limit infection at the site of inoculation (**Fig. 2a**) while increasing JEV penetration into the CNS (**Fig. 2g**). We changed the wording in this heading (p.7, line 118) to more accurately reflect that clearance is enhanced at the inoculation site rather than the entire periphery. As requested, the virus titers in the serum and lymph nodes at various time points have now been assessed and the data added to the manuscript (**Figs. 2b,d**). The liver and kidney are not known to be targets for JEV infection so we did not focus on obtaining data related to these organs. Indeed, when we assessed JEV titer in livers of mice, we did not detect infection (data not shown). The new results we added showing viremia and infection in LNs (**Figs 2b,d**) are consistent with the data we previously provided in the spleen (**Fig. 2c**), showing that MCs do not affect systemic establishment of virus in spite of promoting increased penetration of the CNS (**Fig. 2g**). That we observed a difference in brain virus titers between WT and Sash mice, when viremia levels are the same, provides additional evidence that MCs are important in JEV-induced break down of the BBB.

3. Immune responses in the periphery or CNS such as IFN, antibodies or inflammatory cytokines and chemokines after JEV infection were not analyzed. Increased production of IFN or pro-inflammatory mediators is associated with high brain viral titers and enhanced BBB permeability.

In response to this question, we added data where we measured important cytokines including IFN- γ , TNF, IL-2, IL-4, IL-10, IL-17 and IL-6 (**Fig. 2e-f and S3c-g**). We did not find significant differences between WT and Sash mice for any cytokines (except IL-6 at the early 6 hour time point where Sash mice had higher IL-6 levels, **Fig. S3g**). This supports our conclusion that cytokines are unlikely to be the initiating event for MC-dependent BBB breakdown in JEV infection in WT mice. Instead, we have identified chymase to be important for initiating BBB permeability (**Fig. 5**), and that blocking the MC protease chymase (using both genetic and pharmacological approaches) is sufficient to reduce permeability and infection drastically (**Fig. 5**). We have now included discussion on the topic of cytokines on p.17, line 354-358.

We did not measure antibodies because all mice are experiencing primary infections and vascular leakage is occurring prior to the time point when virus-specific antibodies are detectable. We have previously reported the kinetics of anti-JEV antibody production in mice and the antibodies only are detectable around day 14, which is beyond our experimental time course¹.

4. Experiments including direct inoculation of JEV in the brain will be important to understand the role of MCs in the periphery vs CNS.

We have now provided the requested data in **Fig. S5**. WT mice inoculated *i.c.v.* with JEV had similar levels of virus at early time points, compared to Sash mice, supporting that MCs don't promote infection in the brain, and they even reduce virus at later time points. This supports that MCs contribute to the penetration of JEV into the brain following peripheral inoculation. When the brain is the site of inoculation, MCs promote clearance of virus, similar to the peripheral site of inoculation (**Fig. 2a**).

In addition to addressing the contributions of MCs to virus clearance from the brain, we have also now provided data addressing the role of brain MCs in *permeability* of the BBB. We performed this experiment by selectively ablating CNS and not peripheral MCs by *i.c.v.* inoculation of diphtheria toxin into transgenic *Mcpt5-iDTR* mice. This resulted in depletion of ~80% of brain MCs (**Fig. S8**). Using this depletion strategy, we saw a decrease in BBB breakdown in DT-treated (MC-depleted) compared to vehicle-treated mice (MC-sufficient) mice (**Fig. 3f-g**). Therefore, we conclude that while peripheral MCs initiate JEV penetration of the BBB, CNS MCs further amplify BBB breakdown after CNS infection. We added discussion of these experiments on p.16, line 332-342.

5. In the experiments where virulent strain is used (Figs. 4 and 5), only one time point is shown (Day 5). It is important to show the kinetics similar to Figs. 2 and 3.

We have now added data from additional time points (Days 3 and 5) for the Nakayama JEV experiments (**Fig. S9c-d**). These additional time points demonstrate that findings from Nakayama are consistent with those observed in SA-14-14-2 JEV. Due to safety and feasibility considerations of working with the BSL3 strain, we have provided more time points in general throughout the manuscript for the attenuated strain SA-14-14-2, but we think our data is thorough in validating the findings with the Nakayama strain at key time points. We believe the targeted use of this strain to validate our findings is an advantage of our study and not a limitation.

6. BBB breakdown should be confirmed by a second method such as IgG (heavy and light chains) levels in perfused brains.

We added the requested data (**Fig. 5h**) and appreciate the suggestion. These results are consistent with data using EBD leakage (**Fig. 5f-g**), and show that WT mice have worsened BBB breakdown compared to Sash mice. Furthermore, the chymase inhibitor TY-51469 reverses this break-down in JEV-infected mice (**Fig. 5f-h**), supporting the mechanism for BBB leakage that we identified.

7. Only viral RNA is shown. No infectious virus titer or plaque assay.

Plaque assays have been done for serum of the mice infected with both SA 14-14-2 (**Fig. 2b, S3b**) and Nakayama virus (**Fig. S9b**), showing that the virus isolated from the blood is infectious. Images of the plaques are provided in **Fig. S3b** and **S9b**.

Negative-strand PCR was also performed on brain and spleen samples days 1 and 5 post-infection to confirm JEV replication in vivo (**Fig. S3a**).

Reviewer 2:

1. One major point is the complete lack of human data throughout the manuscript, with the exception of the ROSA cell line experiment. The authors should include an assessment of mast cell activation in JEV patients and should strongly consider addressing whether, similar to dengue and other viruses, mast cells are permissive to JEV. How mast cells become activated by JEV is not addressed in this study and should be considered. If indeed degranulation is dependent on live virus, it would strongly suggest the cells are undergoing infection. Whether infection is fully permissive or abortive will need to be assessed. Perhaps the decrease in JEV titers in Sash mice is at least partly, the result of a key cellular target of infection being lost.

We believe that our study, when published, will be a good justification to examine JEV-infected patients for correlates of MC activation but this would be out of the scope of this project. It may take years to design and obtain funding and approvals for such a large-scale human study. However, we do have data indicating relevance to humans since we have shown that human MCs are also activated by JEV (**Fig. 1h, S2, S9a**). To address the reviewer's question whether MC-degranulation to JEV is replication dependent, we have added data showing that UV-inactivated JEV induces an equivalent degranulation response (**Fig. S2**). Therefore, we can conclude that the response is not replication dependent. To clarify, Sash mice actually showed *higher* levels of virus at the site of inoculation (**Fig. 2a**) and similar levels of infection on a systemic level in the serum, spleen, and lymph nodes (**Fig. 2b-d**), so we do not see any evidence that a key infection target has been lost.

2. The authors make use of two strains of JEV, a clinical isolate and an attenuated strain. Both of these strains need considerably more information. How is the attenuated strain attenuated? Does it still cross the BBB? Or are there issues with it getting into brains as it does not show encephalitis? These differences are important to

understanding the role of the mast cell in the pathogenesis of the virus infection. The authors suggest that mast cells augment or promote BBB breakdown in response to JEV, independent of the strain they used. Both strains show enhanced mast cell activation with degranulation. But only one strain induces disease? How do they reconcile the differences?

We have used well-established attenuated (SA-14-14-2) and virulent (Nakayama) strains of JEV that have been described in numerous publications, many of which we have cited²⁻⁴. This question has now been directly addressed in the Discussion (p. 18, line 373-384), with an explanation that SA 14-14-2 has intrinsic defects in neurovirulence, which is the ability to cause damage within neural tissue during infection⁵. This is independent from the property of neuropenetrance².

3. The authors propose that in the absence of mast cells, using the *Wsh/wsh* MC-deficient mice, that there is reduced neuroinvasiveness by JEV. MCs promoted enhanced infection of the CNS via chymase released during degranulation. If MCs are responsible for this only by modifying the blood brain barrier integrity, inoculating virus directly into the brain should surpass the dependence on the MC for virus-induced symptoms including neurological deficit and survival, that develop after JEV infection of the central nervous system. This is a crucial experiment to ascertain MCs mediate neuropenetrance in JEV.

The reviewer is correct that *i.c.v.* injection of virus directly into the brain bypasses the requirement of MCs for JEV entry into the brain and the WT and *Sash* mice both show severe morbidity/mortality by day 6 post-infection (**Fig. R1**). JEV inoculated *i.c.v.* in WT mice showed similar levels of virus at early time points but increased clearance in WT mice compared to *Sash* mice at late time points (**Fig. S5**). This shows that MCs can promote clearance of virus in an experimental context in the brain. However, in this experimental context where the peripheral blood is not infected, we have bypassed the contributions of MCs to BBB penetration. These results, cumulatively support the role of MCs in BBB breakdown and neuropenetrance, while showing that neurovirulence is dependent upon infection in the brain.

Figure R1: Similar survival for WT and Sash mice after i.c.v. inoculation of JEV

6. In Figure 1. A) The authors need to include an additional larger field of view showing more than just two mast cells. This image should give a better representation of how many mast cells are activated in JEV infection at 5 days.

As requested, we have now included an image with larger field of view (**Fig. S1a**).

7. In figure D and E the authors show an in vitro degranulation response, by IF and by release of beta-hex within 1 hr. Given this accelerated time frame, the use of an appropriate UV-inactivated virus controls need to be included. Does simply binding induce degranulation given that both the virulent and attenuated strains induce this activation?

Degranulation assays using UV-inactivated viruses have been performed. We still observe significant increase in degranulation, indicating that degranulation is not dependent on JEV replication (**Fig. S2**).

Furthermore, A direct comparison, in the beta-hex degranulation assay, as well as by counting of the number of activated degranulated mast cells in brains of mice inoculated with both the clinical and attenuated strains, should be done. Does the clinical strain activate more mast cells? Release more chymase? Do mast cells release other granule contents? Or chymase specifically? The authors need to measure both tryptase and histamine release as these are the two most common mediators released during degranulation. Are they not released? Is that why they were not included? This would be interesting.

Due to differences in the virulence of the strains *in vivo*, different optimal inoculating doses and differing virus replication efficiencies^{2,4}, we don't think a comparison of the

amounts of MC degranulation in the brain between strains can be correctly controlled for strain-to-strain comparison *in vivo*. We believe the most clear quantitative assays for showing that degranulation is similar to both strains of JEV are shown *in vitro* in **Fig. 1e,h, S2, S9a**.

We don't expect some granule products but not others would be released because we are measuring degranulation, which is a process where granule-associated products are released together as pre-formed particles that contain many mediators bound together. To address the reviewer's request for the data showing other granule-associated products are concurrently released, we have now confirmed this is the case since tryptase activity and histamine levels are increased after JEV exposure of MCs (**Fig. S12**). Furthermore, we provide data that the chymase inhibitor is chymase-specific and does not affect other MC mediators (**Fig. S12**).

8. In figure 1H – the authors indicate that activation is not limited to mouse mast cells but also human cells. The ROSA mast cell line is used to address this. This is a relatively new cell line and not widely used. The authors need to confirm the ability of JEV, both the virulent and attenuated strains, to activate human mast cells *in vitro* using either the LAD2 line from the Kirshenbaum Lab or primary cord-blood-derived mast cells as these are both standards in the field.

We have now added data that the Nakayma strain induces degranulation of ROSA cells (**Fig. S9a**), much like in the live-attenuated virus (**Fig. 1h**). The ROSA cell line has many advantages over the LAD2 line, which is well-known to be hypogranulated⁶. Nevertheless, we have confirmed that degranulation of LAD2 cells occurs in response to JEV (**Fig. R2**).

Figure R2: Beta-hexosaminidase assay using LAD2 cells also shows increased MC degranulation by JEV.

9. In figure 2 the authors propose that (2A) attenuated JEV clearance is enhanced in the peritoneum but no difference (2B) in the total burden of virus in the spleen. First, what gene was used for JEV quantification. It is not in the methods. There is also no indication of any controls to ensure virus being detected was *de novo* synthesized and not simply input virus. This needs to be addressed. The authors also state that pUC19 plasmids with JEV insert was used as a standard. Please indicate the slope and efficiency of the reaction in the methods.

The fact that there was an increase of JEV detection from 6 hours to 24 hours *in vivo* suggests that JEV has undergone replication rather than only detecting the input virus (Fig. 2a). Also, we have now performed negative-strand PCR on the spleen and brain tissues at multiple time points to validate that replication of virus occurs (Fig. S3a). The negative-strand is only synthesized during virus replication. The WT mouse model is well established for JEV infection so this is expected⁷. The PCR standard curve is provided as Fig. R3 and we have added additional details to the methods on p. 22-23, beginning line 467. The standard curve equation is also provided below in Fig. R3. We thought it would be distracting to include this level of experimental detail in the manuscript but are happy to include it if required.

Figure R3: Standard curve and equation used to calculate virus titer from qPCR.

10. The data do not support enhanced clearance as the authors state in lines 117-118. The differences observed in these two expts could be explained by the dramatically reduced number of cells in total in the peritoneum, as mast cells are a major component of this site, thereby increasing the relative amount of JEV in population. To control for this the authors need to determine the composition of the peritoneum during WT and MC-deficient infections. A more appropriate control is to express levels of JEV per mg of tissues as is seen in the figure B.

We think the reviewer may have misunderstood because we cannot express the levels of JEV per mg because the cells of the peritoneum are harvested by peritoneal lavage in a liquid suspension (Fig. 2a). We do not recover significantly different numbers of cells from the peritoneums of WT and Sash mice, probably because they are a minor population of cells. While the peritoneum of a healthy mouse has $\sim 4 \times 10^6$ macrophages, there are only $\sim 6 \times 10^3$ MCs⁸. Most of the cells in the peritoneal cavity are lymphocytes and macrophages⁹, the latter of which are prime targets of DENV infection¹⁰. We think the data are appropriately presented with the optimal methods that are feasible in this case. Also, the difference in relative titer is a 3-log difference, which is highly significant, and can't be accounted for only by the denominator used in normalization (the cell number).

11. With regard to analysis and presentation of data in figure 2 -The the authors note that these were done with a n=6. It is not useful to show the mean? And SEM? (what it is

exactly?? It is not given in the figure legend), given the large variation seen. In the case of the cerebellum, brain stem and spinal cord the variation is such that the data overlaps and is not significant. The only significance is in the cortex and sub-cortex areas. The text, as written, indicate all are significantly different. This is not correct. These figures need to show each data point individually to give a more accurate visual representation of the data with use of asterisk over each bar to show individual significance. Is it that 3 mice showed increased levels and 3 did not? Etc...

We have modified the representation of figures to appear as dot plots, throughout the manuscript, where possible. Specifically, for **Fig. 2g**, we have left the main figure as-is so that the trends are visible but also provided separate graphs with the dot plots for each brain region (**Fig. S4**), as requested. We have also validated that the significant comparisons are correctly indicated, along with the appropriate statistical tests. For **Figure 2**, the differences are significantly different by ANOVA comparing WT versus Sash mice, so the comparison is correctly indicated as significant. We have verified that figure legends contain the information that the means are presented with SEM unless otherwise noted.

12. Lines 127-129 The conclusions here are not supported by the data.

The wording has been changed to the following: "These results indicate that the protective effects of MCs were confined to the initial site of inoculation and similar infection and inflammatory profiles were observed on a systemic level." (p.8, lines 141-145).

13. In figure 3, the authors aim to show mast cells mediate BBB breakdown. In these experiments the authors again use the attenuated JEV strain and show nicely that WT mice demonstrate enhanced loss of integrity. The Evans blue experiment is very nice. The authors note however that at day 5 and day 7 post-infection that the Sash mice did not exhibit leakage as evidence by the Evans blue staining, line 140. This is clearly not correct. The text states the WT mice exhibited worsened BBB breakdown, lines 136-138. The Evans blue images also clearly show a level of leakage that is less than observed in WT mice, but still present.

The wording has been changed to the following: "This could be visualized in brains where, in contrast to WT mice which showed extensive leakage of the blue-colored dye, leakage was visibly reduced in Sash mice (**Figure 3b**).” The quantitative data in the form of graphs supporting this assessment are presented in **Fig. 3a,d,f**.

13. In figure 3A, showing quantification needs to show individual points, not the mean and SEM?? of the six mice used, if this is what is plotted? Asterisks need to be placed on the graph, over the appropriate days that show differences. Is Day 7 significant? Or day 5? Or day 3?

The graphs in **Fig. 3a** have been modified to clarify the days that showed significant differences. We also included a dot-plot format in **Fig. S6** that corresponds to this graph.

14. In part C of figure 3 the authors note no change in Zo-1 levels in Sash mice infected with JEV. The IF images indicate a significant upregulation in intensity of Zo-1 staining. The authors make no mention of this. Please explain.

The authors could not agree that ZO-1 appears upregulated. Since the observation is not consistent in our quantitative data, we don't want to make the claim of upregulated ZO-1 staining. In contrast, the downregulation of ZO-1 staining that we reported was very consistent (**Fig. 3c**) and supported by other techniques including western blotting (**Fig. 5e**).

15. In figure 4, the authors look at the mast cell involvement in virulent infections and indicate that WT mice fair far worse than MC-deficient mice in viral titer, body weight, grip strength and survival. Again the authors show increased viral titer in various areas of the brain in the presence of mast cells. The legend indicate an n=8. Again Each point must be shown to see the true distribution of the data.

Dot-plots were added, as requested.

16. Figures 4 E and F are very hard to read. The circles, which I assume are individual mice (this should be clearly indicated in the legend) are difficult to see. Additionally, if both measurements D and E were made on the same group of 5 mice then each mouse must have it's own symbol so that they can be followed both over time and between symptoms. If repeated measuring of one mouse was done for each time point then you must ensure that individual responses can be tracked. le did the one mouse that exhibited a decrease in body weight also show the decreased grip strength? Does the same mouse worsen over time? Etc.. The authors also need to describe what is happening to the WT and Sash mice that appear to fall off the curve? Do they die? There is no mention of them at all.

We have now edited the figure legend to indicate that individual mice are shown. We didn't think giving each mouse its own symbol was required here because the lines connect the points. As requested, we have also alternatively represented this as the averages of n=15 mice in **Figure S10**. Symbols were added to indicate mice that died and were no longer represented on the graphs.

17. Figure 4 F needs error bars!

We have added the 95% confidence intervals to the Kaplan-Meier curves in the figure.

18. In figure 5 the authors nicely demonstrate that MCs and JEV induce permeability and that blocking or inhibiting chymase with the small molecule inhibitor TY-51469 abrogates this effect. The effect was found to be a decrease in the tight junction proteins Claudin-5, Zo-1 and Zo-2 and occludin. This did not occur when tryptase was blocked. Given the potential for off target effects with inhibitors the authors should confirm that it is chymase that mediates these effects using MC-chymase deficient mice. These models exist and are available for use. Additionally, crispr/cas knockouts would easily address this question. The authors make the statement that chymase induces the BBB permeability by modifying tight junctions but only show the dependence on chymase specifically with one small molecule inhibitor.

As requested, we now used bone marrow from chymase-deficient mice to address this question. Similar results were obtained to the chymase inhibition data that JEV activation of BMMCs lacking chymase does not reduce the permeability of endothelial monolayers (**Fig. 5c-d**). This data was added to the manuscript. We have also provided data

showing the specificity of the drug TY-51469 to chymase, in that it does not influence the release or activity of other mediators including histamine or tryptase (**Fig. S12b-c**).

If chymase is indeed the key mediator that breaks down the BBB, then addition of enzymatically active chymase (see R&D Systems or Sigma) would facilitate enhanced JEV infection, symptoms and mortality in Sash mice, in the absence of the MC themselves. This experiment needs to be included to make the conclusion.

We have performed the requested experiment and, indeed, as reviewer predicts, injection of exogenous chymase enhances JEV penetration of the BBB *in vivo* (**Fig. 5j-k**).

19. In figure 6, the authors use their mouse models to show treatment with the small molecule inhibitor abrogates disease. Again, the addition of a model that specifically lacks chymase needs to be used along with the experiment that administers active recombinant chymase to recapitulate the effect. With many lines of evidence the chymase dependence will be considerably more convincing.

As mentioned above, we have now provided data that administration of exogenous chymase in MC-deficient, JEV-infected mice is sufficient to recapitulate the phenotype of increased vascular leakage and enhanced infection (**Fig. 5j-k**). Furthermore, we have provided data using BMMCs that lack the mouse chymase MCPT4, which also support the phenotype is chymase-dependent (**Fig. 5c-d**).

20. Figure 6 C – ERROR bars!

95% confidence intervals have been added to the survival curve (**Fig. 6c**).

21. The authors need to address the issue of both the attenuated strain and clinical strain induce degranulation and chymase release, and both strains show enhanced JEV titers in brain when MCs are present, but not when they are absent. How do the authors explain the lack of disease in the attenuated strains, even though they see the same MC activation events and chymase release and virus in the brains. Is it simply a numbers game and there is less virus there so less disease? This needs to be reconciled. If this is true, then the experiment to address neuropenetrance needs to be done.

SA-14-14-2 virus has been shown to have less *neurovirulence* compared to other virulent strains such as Nakayama^{2,11}. This means that although it can infect the brain, it is not able to replicate as efficiently in neurons as wild-type strains. Indeed, injection of the brain directly with SA-14-14-2 does not lead to mortality while injection of NKYM does². Neurovirulence is an unrelated mechanism compared to neuropenetrance. An advantage of our data is that we have shown with both virulent and attenuated strains that neuropenetrance can be affected by MCs. Neuropenetrance is also key to the survival of the animals, since we can block the penetrance of the fully virulent strain and increase survival (**Fig. 6c**). In our study, we have identified a host factor, chymase, that is important for neuropenetrance. While neurovirulence, which is also important for pathogenesis, is virus-intrinsic and beyond the scope of this study. This was an important point the reviewer raised and we have now added discussion to clarify (p.18, lines 373-384).

Methods points

The authors should ensure by cell authentication that the lines used were not contaminated with other lines. They should also indicate the status of mycoplasma in the cultures and describe the protocol and testing schedule used to ensure lines were all mycoplasma free.

All cell lines other than ROSA cells were obtained directly from ATCC and were mycoplasma negative. ROSA cells were tested for mycoplasma and also were negative. The methods were updated to indicate this (p.20-21, lines 428-438).

The authors need to indicate which viral protein is being amplified by qPCR and what steps were taken to ensure there was no contamination for input virus.

This information was added to the manuscript methods p.22-23, lines 468-498). In brief, negative control (uninfected) tissues were run with each assay to ensure specific amplification without contamination.

The TEER assay needs more details. How many cells were plated? In what volume? What Passage of cells were used? How long did they leave to form a completely impermeable confluent monolayer? What electrode was used in the measurement? What material was used in the transwell inserts.

This information was added to methods (p.25, lines 528-542).

Minor points-

Line 28 – define immune privilege and provide a reference

“Immune privilege” is a standard concept so we have not defined it here but added a review citation that discusses it in depth (p.3, line 30, citation 10)

Lines 30-32 – Is this in reference to human studies? If so you should reference. If in reference to the mouse study, then you should combine sentences.

This was in reference to the mouse study, as stated in the previous sentence. The sentences have been combined to clarify (p.3, line 32-34).

Line 52-54. Is this in mice? Or humans? Needs to be clarified.

This was clarified (p.4, line 53-55).

Line 332 – ROSA cell line, Indicate the institution for Arock

Affiliation has been included in the acknowledgements (p.31, line 584).

Line 338 – indicate the institution for Ooi.

Affiliation has been included in the acknowledgements (p.31, line 585).

Reviewer #1 (Remarks to the Author):

The authors have addressed all my concerns. The manuscript is technically sound and fulfills the criteria for publication.

Reviewer #2 (Remarks to the Author):

The authors have addressed all the comments.

We are grateful for the recommendation of the reviewers to accept our manuscript.

Citations

- 1 Saron, W. A. A. *et al.* Flavivirus serocomplex cross-reactive immunity is protective by activating heterologous memory CD4 T cells. *Sci Adv* **4**, eaar4297, doi:10.1126/sciadv.aar4297 (2018).
- 2 Chambers, T. J., Droll, D. A., Jiang, X., Wold, W. S. & Nickells, J. A. JE Nakayama/JE SA14-14-2 virus structural region intertypic viruses: biological properties in the mouse model of neuroinvasive disease. *Virology* **366**, 51-61, doi:10.1016/j.virol.2007.04.016 (2007).
- 3 Arroyo, J. *et al.* Molecular basis for attenuation of neurovirulence of a yellow fever Virus/Japanese encephalitis virus chimera vaccine (ChimeriVax-JE). *Journal of virology* **75**, 934-942, doi:10.1128/JVI.75.2.934-942.2001 (2001).
- 4 Chambers, T. J., Nestorowicz, A., Mason, P. W. & Rice, C. M. Yellow fever/Japanese encephalitis chimeric viruses: construction and biological properties. *Journal of virology* **73**, 3095-3101 (1999).
- 5 Yang, D. *et al.* Characterization of live-attenuated Japanese encephalitis vaccine virus SA14-14-2. *Vaccine* **32**, 2675-2681, doi:10.1016/j.vaccine.2014.03.074 (2014).
- 6 Guhl, S., Babina, M., Neou, A., Zuberbier, T. & Artuc, M. Mast cell lines HMC-1 and LAD2 in comparison with mature human skin mast cells--drastically reduced levels of tryptase and chymase in mast cell lines. *Exp Dermatol* **19**, 845-847, doi:10.1111/j.1600-0625.2010.01103.x (2010).
- 7 Li, F. *et al.* Viral Infection of the Central Nervous System and Neuroinflammation Precede Blood-Brain Barrier Disruption during Japanese Encephalitis Virus Infection. *Journal of virology* **89**, 5602-5614, doi:10.1128/JVI.00143-15 (2015).
- 8 Ajuebor, M. N. *et al.* Role of resident peritoneal macrophages and mast cells in chemokine production and neutrophil migration in acute inflammation: evidence for an inhibitory loop involving endogenous IL-10. *J Immunol* **162**, 1685-1691 (1999).
- 9 Ray, A. & Dittel, B. N. Isolation of mouse peritoneal cavity cells. *J Vis Exp*, doi:10.3791/1488 (2010).
- 10 Schmid, M. A. & Harris, E. Monocyte recruitment to the dermis and differentiation to dendritic cells increases the targets for dengue virus replication. *PLoS Pathog* **10**, e1004541, doi:10.1371/journal.ppat.1004541 (2014).
- 11 Gromowski, G. D., Firestone, C. Y. & Whitehead, S. S. Genetic Determinants of Japanese Encephalitis Virus Vaccine Strain SA14-14-2 That Govern Attenuation of Virulence in Mice. *Journal of virology* **89**, 6328-6337, doi:10.1128/JVI.00219-15 (2015).